# Multiple ancestral haplotypes harboring regulatory mutations cumulatively contribute to a QTL affecting chicken growth traits

Yuzhe Wang [1,2,7], Xuemin Cao [1,7], Chenglong Luo[3], Zheya Sheng[1,4], Chunyuan Zhang[1,5], Cheng Bian [1], Chungang Feng[1], Jinxiu Li[1], Fei Gao[1,2,5], Yiqiang Zhao[1,5], Ziqin Jiang[1], Hao Qu[3], Dingming Shu[3✉], Örjan Carlborg[6✉], Xiaoxiang Hu [1✉] & Ning Li[1]

In depth studies of quantitative trait loci (QTL) can provide insights to the genetic architectures of complex traits. A major effect QTL at the distal end of chicken chromosome 1 has been associated with growth traits in multiple populations. This locus was fine-mapped in a fifteen-generation chicken advanced intercross population including 1119 birds and explored in further detail using 222 sequenced genomes from 10 high/low body weight chicken stocks. We detected this QTL that, in total, contributed 14.4% of the genetic variance for growth. Further, nine mosaic precise intervals (Kb level) which contain ancestral regulatory variants were fine-mapped and we chose one of them to demonstrate the key regulatory role in the duodenum. This is the first study to break down the detail genetic architectures for the well-known QTL in chicken and provides a good example of the fine-mapping of various of quantitative traits in any species.

[1] State Key Laboratory of Agrobiotechnology, College of Biological Sciences, China Agricultural University, Beijing 100193, China. [2] College of Animal Science and Technology, China Agricultural University, Beijing 100193, China. [3] State Key Laboratory of Livestock and Poultry Breeding, Guangdong Key Laboratory of Animal Breeding and Nutrition, Institute of Animal Science, Guangdong Academy of Agricultural Sciences, Guangzhou 510640, China. [4] Key Laboratory of Agricultural Animal Genetics, Breeding and Reproduction of Ministry of Education, College of Animal Science and Technology, Huazhong Agricultural University, Wuhan 430070, China. [5] Beijing Advanced Innovation Center for Food Nutrition and Human Health, China Agricultural University, Beijing 100193, China. [6] Department of Medical Biochemistry and Microbiology, Uppsala University, Uppsala SE-751 23, Sweden. [7] These authors contributed equally: Yuzhe Wang, Xuemin Cao. ✉email: shudm@263.net; orjan.carlborg@imbim.uu.se; huxx@cau.edu.cn

The molecular mechanisms of many monogenic traits have been identified in agricultural animals and plants, facilitated by their distinct genotype to phenotype associations[1–3]. In contrast, complex quantitative traits tend to follow Fisher's[4] minor-polygene hypothesis, and are difficult to disentangle due to small allelic effects for each locus, complex patterns of linkage disequilibrium in associated regions, limited ability to screen for putative functional polymorphisms, the balance between artificial and natural selection for complex traits, and variable effects of loci across populations and environments[5–7]. In human genetics, genetic variants contributing to complex traits such as human height are often distributed across the entire genome and an infinitesimal model seems to hold[8]. In domestic animals, although some prominent examples including *Insulin-like growth factor 2* (*IGF2*)[9] and *Myostatin* (*MSTN*)[10,11] prove that artificial selection has resulted in fixation of large effect mutations, the on-going grand challenge in the field is still to provide a better empirical understanding of the basis for polygenic traits.

A major-effect, quantitative trait locus (QTL) has been mapped in many chicken populations at the distal end of chromosome 1, where it has been confirmed to be significantly associated with several growth-related traits including body weight, abdominal fat, muscle weight, and feed conversion ratio[12–21]. Many candidate genes reside in the associated region, including *Retinoblastoma 1* (*RB1*), *Forkhead box O1* (*FOXO1*)[20,21] and an insertion mutation of *miR-16* have been suggested as possible causal polymorphism[15]. However, there is still no conclusive consensus among the detailed genetic architectural, biological mechanisms of this locus. In our previous study, an F2 intercross between a slow growing meat type chicken, the Huiyang Bearded (HB, Table 1), and a fast-growing commercial broiler breed, High Quality chicken Line A03 (HQLA, Table 1) were constructed. We also mapped this QTL for growth-related traits in this region to the distal end of GGA1[18].

Here, we report the construction of an F15 advanced intercross line (AIL) derived from the previously reported F2 population and we use the large F9 population for QTL mapping and birds from the F15 generation for functional studies. The strongest association was detected to a mosaic haplotype that was further fine-mapped and functionally explored by evaluating the relationship between the haplotype detected in the HQLA-HB population and in two independently selected broiler breeds, six ancient low body weight breeds and two jungle fowl populations (Table 1)[22,23]. Whole genome sequencing revealed multiple ancestral haplotypes harboring regulatory mutations cumulatively contributing to the major QTL affecting chicken growth traits. We further identified one instance where sequence variants within a regulatory element was identified as candidate causal mutations regulating the expression of the *calcium binding protein 39 like* (*CAB39L*) gene. This study reports the fine-mapping and functional dissection of a complex quantitative trait locus, contributing new insights to the genetic and biological mechanisms underlying growth in chicken.

## Results
### Genome-wide association studies replicate a major growth trait QTL on GGA1.
Genome-wide association study (GWAS) analyses for several body weight and carcass traits (Supplementary Tables 1 and 2) were performed separately in the F2 and F9 generations from the HQLA × HB intercross population using in total 41,758 and 291,772 SNPs[24], respectively. In the F2 generation ($n = 493$), a single, genome-wide 1% significant ($p = 8.1 \times 10^{-7}$) association signal was detected for 11 of 15 evaluated traits at the distal end of GGA1 (165 Mb to 175 Mb; Supplementary

Table 3; Supplementary Fig. 1). The most significant association was to a SNP (rs14917305) located at 169,795,686 bp ($p = 1.6 \times 10^{-18}$) for the combined weight of the ventriculus and the proventriculus (SW; Table 2 and Supplementary Fig. 2). For body weight, the most significant associations were for weight at 10 weeks of age (BW10; GGaluGA055630 at 171,387,660 bp; $p = 1.2 \times 10^{-10}$; Table 2 and Supplementary Fig. 1), and weight at 8 weeks of age (BW8; GGaluGA055630; $p = 2.1 \times 10^{-9}$; Fig. 1a).

In the F9 generation ($n = 595$ after quality control from 602 samples), the strongest association was to BW8 (S1_168536487 at GGA1: 169,241,142 bp; $p = 3.4 \times 10^{-16}$; Fig. 1b; Table 2 and Supplementary Table 4). The QTL peak was narrower in the F9 than in the F2 population due to the additional recombinants and higher density of SNP markers (41 K vs. 292 K). The top SNP (S1_168536487) was significantly associated with 10 traits, including weight at 2–14 weeks of age, growth rates (0–4 and 4–8 weeks of age) and intestinal length (IL) (Table 2 and Supplementary Table 4). To further define the associated haplotype around S1_168536487, SNPs were aggregated using $r^2 = 0.3$ with the top SNP as a criterion. This identified a target region of 3.1 Mb from 168.6 Mb to 171.7 Mb (Fig. 1c). In addition to the GGA1 QTL, a second QTL was discovered for BW8 in F9 on GGA27 (3.60 Mb to 3.75 Mb; S27_3406188 at 3,620,306 bp; $p = 7.6 \times 10^{-6}$), containing genes including *Insulin-like growth factor 2 binding protein 1* (*IGF2BP1*) and *Phosphoethanolamine/phosphocholine phosphatase* (*PHOSPHO1*).

### Identifying breed-level recombinants in the 3.1Mb region on GGA1.
The GWAS *P*-values and the difference in allele-frequency between HQLA and HB at the significant SNPs (ΔAF (HQLA-HB) were highly correlated ($r = 0.68$, Supplementary Fig. 3). This correlation is consistent with the basic assumption in our analyses that, at many loci, alleles with significantly different effects on growth (growth-increasing Q in HQLA and growth-decreasing q in HB) were present in considerably different frequencies in these two phenotypically divergent populations. To trace haplotypes inherited from the founder breeds through the experimental cross, we first used founder-discriminatory markers (breed-level) to perform an identical-by-descent (IBD) analysis from F0 to F9. Only 2 of the ~800 GBS markers in the selected 3.1 Mb region (168.6 Mb–171.7 Mb; Fig. 1c) were informative of founder-breed origin (defined as having delta allele-frequency differences between the HQLA and HB, $|\Delta AF_{(HQLA-HB)}| > 0.95$). Therefore, we sequenced the 31 F0 founders of the AIL to 10× individual coverage to identify additional markers for tracing the recombination events between the HQLA and HB founder haplotypes in the associated GGA1 QTL region at high resolution. The 3.1 Mb peak QTL region was screened for polymorphic sites in the sequence data (Supplementary Table 5) and in total 46 SNP polymorphisms with $|\Delta AF_{(HQLA-HB)}| > 0.95$ were identified. Next, 31 of these 46 SNPs were selected for genotyping using Fluidigm® in from the F9 generation ($n = 575$ after quality control from 602 samples). An IBD analysis identified four breed-level recombination breakpoints in the QTL interval. In total, 503 F9 individuals carried unrecombined progenitor chromosomes (HQLA/HQLA, HQLA/HB, HB/HB) in the 3.1 Mb region and significant differences in BW8 were detected between individuals carrying the three different founder-breed origin genotypes (Fig. 1d). The other 72 F9 individuals carried one or two copies of the recombinant haplotypes (the different recombinant haplotypes found among these 72 individuals are shown in Fig. 1e). An association analysis to the IBD-status of the five blocks defined by the four recombination events found that only the first two proximal blocks were significantly associated with BW8 ($p = 0.0007$ and 0.011, respectively, Fig. 1e) and only the most

**Table 1 Summary of phenotype and sequencing information for the breeds included in the study.**

| Cluster | Breeds or lines | Abbreviation | Distribution | Total sample # | Sequencing depth | Body weight at 7 weeks of age (g) | Ref. |
|---|---|---|---|---|---|---|---|
| Advanced intercross lines (AIL) | High quality line A03 | HQLA (F0) | Commercial, China | 16 | 10× | 1751.9 ± 173.1 | This study |
| | Huiyang Bearded chicken | HB (F0) | Guangdong, China | 15 | 10× | 545.3 ± 55.2 | This study |
| | $F_2$ intercross (HQLA × HB) | F2 | Guangdong, China | 493 | -(Chip) | 954.5 ± 158.5 | 18 |
| | $F_9$ AIL (HQLA × HB) | F9 | Guangdong, China | 595 | 10× (GBS) | 966.6 ± 149.3 | 24 |
| Commercial broiler (high body weight) | Cornish | CB1 | Commercial, America | 29 | 8× | 3282.4 ± 398.7 | This study |
| | White Plymouth Rock | CB2 | Commercial, Europe | 30 | 8× | 1247.5 ± 108.2 | This study |
| Native breed (low body weight) | Silkies | SK | Jiangxi, China | 30 | 8× | 463.4 ± 70.4 | This study |
| | Chahua chicken | CH | Yunnan, China | 30 | 8× | 400.0 ± N.A. | This study |
| | Tibetan chicken | TBC | Tibet, China | 30 | 8× | 346.1 ± 34.8 | This study |
| | Daweishan Mini chicken | DWS | Yunnan, China | 31 | 8× | 295.4 ± 39.8 | This study |
| | Xishuangbanna Game fowl | XSD | Yunnan, China | 8 | 26× | 289.0 ± 24.5 | 22 |
| | Sumatera and Kedu Hitam | SUM | Indonesia | 15 | 8× | N.A. | 23 |
| Jungle fowl (low body weight) | Red jungle fowl | RJF | Indonesia | 10 | 10× | N.A. | 22,23 |
| | Green jungle fowl | GJF | Indonesia | 9 | 10× | N.A. | 23 |

**Table 2 Top SNPs for each trait in $F_2$ and $F_9$ generations.**

| Top SNPs | Chr | Location | Population | AF(HQLA) | AF(HB) | △AF | Traits |
|---|---|---|---|---|---|---|---|
| S1_168536487 | 1 | 169,241,142 | $F_9$ | 1.00 | 0.03 | 0.97 | BW2, BW4, BW6, BW8, BW10, BW12, BW14, IL |
| rs13974906 | 1 | 171,778,431 | $F_2$ | 0.16 | 1.00 | 0.84 | BW2, BW4, GR0–4 |
| GGaluGA055431 | 1 | 170,802,565 | $F_2$ | 0.88 | 0.13 | 0.74 | BW6 |
| GGaluGA055630 | 1 | 171,387,660 | $F_2$ | 0.84 | 0.00 | 0.84 | BW8, BW10, BW13 |
| GGaluGA054960 | 1 | 169,679,782 | $F_2$ | 0.28 | 1.00 | 0.72 | BW12 |
| rs14917305 | 1 | 169,795,686 | $F_2$ | 0.09 | 0.97 | 0.87 | SW |
| GGaluGA054496 | 1 | 168,446,325 | $F_2$ | 0.88 | 0.00 | 0.88 | IL |

proximal block was significantly associated with BW14 ($p = 0.0009$, Supplementary Fig. 4). For each block interval, we also computed the detailed phenotypic scales for the three genotypes and significant differences were observed among the three groups only in the first two proximal blocks ($p < 0.05$, Fig. 1f). The differences between Block2 and Block3 were due to 28 HQLA to HB conversions (Supplementary Fig. 5), resulting in a significantly lower BW8 of HQLA/HB and HB/HB individuals in Block1 and Block2 compared to Block3, Block4, and Block5 (HQLA/HQLA not included due to small sample size, Fig. 1f). Besides, this interval (Block1+Block2) overlapped with the peak QTL region in other mapping studies[12,15,21] and the most significant SNP in our genome-wide analysis was located in this region. In total, this segment tagged by GBS markers explained 14.4% of the genetic variance in BW8 in the $F_9$ generation of the AIL. Conservatively, we selected the ~1.2 Mb region covered by Block1 and Block2 (168.6–169.8 Mb) for further analysis.

**Identifying a mosaic pattern of the 1.2 Mb QTL interval.** In an attempt to utilize individual-level haplotype diversity in the founders to fine-map the region further, additional genotyping of markers segregating in the two intercrossed founder populations

was performed across the region. Only 7 of the 31 genotyped founder-population informative SNPs ($|\Delta AF_{(HQLA-HB)}| > 0.95$) were located in the 1.2 Mb candidate region. For this, an additional 76 SNPs with $|\Delta AF_{(HQLA-HB)}| > 0.75$, including the peak SNP S1_168536487 in the previous $F_9$ GWAS, were selected in this interval (detail in "Methods" section). These were genotyped in the same $F_9$ cohort as before for use in additional association analysis and exploration of common, segregating haplotypes. Of these, 60 SNPs were selected as TagSNPs using a 1% genome-wide Bonferroni corrected significance threshold in the single SNP $F_9$ association analysis across these markers. Nine of these SNPs were more significant in this analysis than the earlier top-SNP S1_168536487 (Supplementary Data 1). When evaluating the distribution of the associated SNPs in the target region in more detail, it was noticed that the TagSNPs were not continuously distributed across the region. They were instead located in clusters separated by regions with SNPs showing low or no association to growth (Fig. 1g). It is considered unlikely that the non-continuous association signal observed was due to a recombination breakup of founder haplotypes in the AIL due to the limited number of generations of intercrossing ($F_0$ to $F_9$) and the recombination rate in chicken. Instead, it is considered more likely to have arisen from variable allele frequencies of the SNPs in the founders

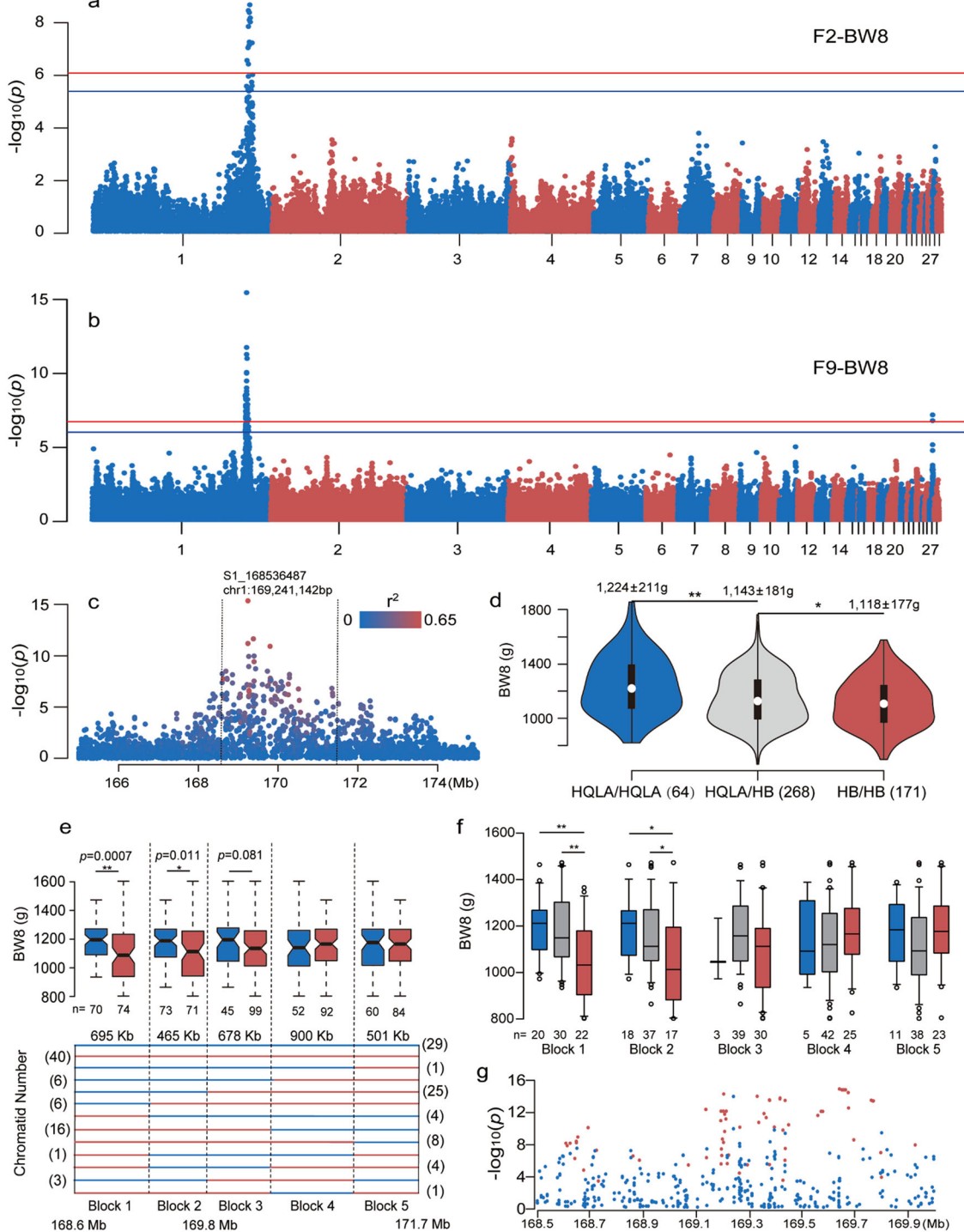

**Fig. 1 Fine mapping of a growth and body composition QTL on GGA1.** Manhattan plots for associations to body weight at 8 weeks of age (BW8) in the $F_2$ (**a**) and $F_9$ (**b**) generations of the HQLA-HB deep intercross population. The horizontal red/blue lines indicate the genome-wide 1/5% significance thresholds (P values = $8.1 \times 10^{-7}$ in $F_2$ and $1.8 \times 10^{-7}$ in $F_9$)/(P values = $4.1 \times 10^{-6}$ in $F_2$ and $9.2 \times 10^{-7}$ in $F_9$), respectively. **c** Scatter plot illustrating all tested SNPs in the QTL region on GGA1 for BW8 in the $F_9$ generation. The colors of the dots indicate their LD ($r^2$) with the peak SNP (GGA1 at 169,241,142 bp, the highest red dot). **d** The detailed phenotypic scales (mean ± SD) for the three genotypes (HQLA/HQLA, HQLA/HB, and HB/HB) in the 168.6–171.7 Mb region in 503 individuals carrying unrecombined progenitor chromosomes. **≤0.01 significance level; *≤0.05 significance level. **e** Seventy-two individuals (144 chromatids) in the $F_9$ generation carry one or two recombinant $F_0$ chromosomes in the peak GGA1 region from 168.6–171.7 Mb. Blue/red lines represent chromosome segments inherited from the HQLA/HB and a block-wise association analysis across this region identified a significant association only to block1 and block2 (168.6–169.8 Mb) to BW8 (P = 0.0007/0.011/0.081/0.326/0.429, respectively). Sample sizes are given below the box plots. Numbers in brackets represent corresponding number of recombinant chromatids. **f** The detailed phenotypic scales for the three genotypes (HQLA/HQLA, HQLA/HB, and HB/HB) for each block in 72 individuals carrying recombined progenitor chromosomes. **g** Regional association plots for the significance testing to the 76 additionally genotyped SNPs (red) and GBS-SNPs (blue) across the 1.2 Mb window from 168.6–169.8 Mb.

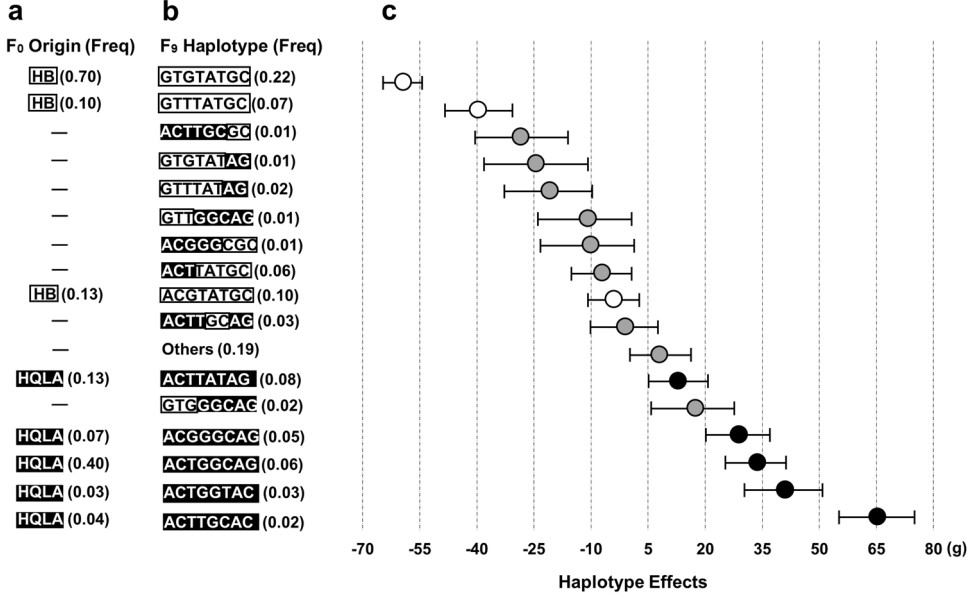

**Fig. 2 Haplotype association analysis for 8-week body weight in the 1.2 Mb candidate region on GGA1.** Eight SNPs were associated with 8-week body-weight in a multi-locus backward-elimination analysis across the 1.2 Mb segment. Haplotypes were estimated in the founder populations (HB and HQLA) and the $F_9$ AIL generation across these markers. **a** Eight haplotypes were inferred in HB ($n_{hap} = 3$) and HQLA ($n_{hap} = 5$) at haplotype frequencies (HF) > 0.01. **b** Unrecombined founder haplotypes segregated at 0.02 < HF < 0.22 in the $F_9$ generation. Eight additional recombinant haplotypes were detected at HF > 0.01 in the $F_9$ generation, and the joint frequency for the group of rare (HF < 0.01; $n = 84$) haplotypes was 0.19. **c** The haplotype substitution effects of the 16 major haplotypes (eight unrecombined $F_0$ and eight recombined) and the "other" group exhibited a gradual distribution of effects on BW8 in the $F_9$ with the unrecombined HB and HQLA haplotypes clustering at low/high effects, respectively. Intermediate effects were observed for recombined segments and one haplotype from each HB and HQLA. This indicates that the haplotype effects were due to polymorphisms at more than one locus in the segment as complete segments from each founder was needed to display the full effect on weight, and also that both HB and HQLA were still polymorphic at one or both of these loci.

(Supplementary Fig. 6). No strong positive selection signal was found using standard measures of the genetic diversity (ΔAF/π/haplotype diversity (H)/Fst/XP-EHH; Supplementary Figs. 6, 7, 8, and 9). These results are consistent with the breeding history of the commercial HQLA stock (details in Materials section), that was formed by first crossing two divergent but outbred population (increasing the haplotype diversity) and then followed by strong directional selection for increased weight over 10 generations. This likely resulted in selection signatures due to selection of longer old haplotypes than creation of new via genetic hitchhiking across multiple individually contributing SNPs at candidate functional loci. This distinct mosaic pattern observed is therefore different from the genetic architecture expected around a strongly selected single causative mutation, making it difficult to exclude any significant SNPs as tags of contributing variants. Thus, we postulated the hypothesis that this 1.2 Mb region might contain multiple linked functional polymorphisms in the founders, which are to a large extent co-inherited through the pedigree causing the extended association signal.

**Haplotype analysis suggests multiple contributing loci in this region**. To identify the haplotypes contributing to the association signal, a multilocus backward-elimination analysis was performed across the 60 tag SNPs in the 1.2 Mb region[25,26]. In addition, 18 markers outside the QTL with associations in the $F_9$-GWAS ($p < 0.0001$) were selected to control for genetic background effects. In total, this analysis identified 8 SNPs (backward-elimination SNPs, BESNPs, Supplementary Data 1) in the 1.2 Mb region with statistically independent associations to BW8 at a 5% False Discovery Rate (FDR) threshold. Next, the haplotypes tagged by these 8 BESNPs were estimated in the $F_9$ population and, in total, 100 haplotypes were detected of which 16 existed at a

frequency > 0.01 ("Major haplotypes"). The remaining 84 were grouped ("Other haplotypes"; total frequency = 0.19).

The additive haplotype substitution effects on BW8 were estimated, and there was a gradient distribution of haplotype allele effects from decreasing BW8 by 59 g to increasing it by 65 g (Fig. 2). The founder haplotypes with HB and HQLA origin have effects at the opposite ends of the effect spectrum. Two non-recombined founder haplotypes, one from HB ($AF_{HB} = 0.70$) and one from HQLA ($AF_{HQLA} = 0.04$), had bigger weight decreasing/increasing effects, respectively, than the other founder line haplotypes. The recombinant haplotypes had intermediate substitution effects to those of the contributing founder haplotypes. Inheriting a partial founder-line haplotype segment is thus not sufficient to reach their full weight increasing/decreasing effects, respectively. The part of the 1.2 Mb segment transmitted to a recombinant haplotype appears to matter. Inheriting a "HB" segment in proximal part of the haplotype provides most of the "low-weight" effect, with only a smaller contribution by the "HB" segment in the distal part. In contrast, the "high-weight" effect appears to be more evenly contributed by the corresponding "HQLA" segments in the proximal and distal parts. The results from the association analyses therefore lead us to propose the hypothesis that this 1.2 Mb segment contain at least two loci contributing to the differences in weight between HB and HQLA, and that both founder lines are likely to segregate for at least two haplotypes across this region with different effects on weight.

**Haplotype-sharing discovered 9 sub-haplotypes as causal candidates**. A haplotype-sharing analysis across multiple high-weight and low-weight chicken breeds was next used to refine the 1.2 Mb region further. We here assumed that one or more of the multiple favorable regions of the large-effect, multi-locus QTL haplotype

identified in the cross between HB × HQLA (the "Q-haplotype") would also have been selected in other high-weight breeds and that these breeds would therefore, at least in parts, share the Q-haplotype in the region. However, this analysis is also a potential source of false negative variant discrimination. The 253 re-sequenced individuals were divided into two groups (high and low body weight, Table 1, Supplementary Figs. 10 and 11). In the 1.2 Mb segment, nine sub-haplotypes were identified as having large frequency differences ($\Delta AF(Q) > 0.4$) between the high and low-weight breeds (Fig. 3a). The low level of pairwise LD across these segments in the multi-breeds analysis (Supplementary Fig. 12), together with the difference in haplotype frequencies

between high and low breeds, suggests that multiple haplotypes that are mosaics of these sub-segments have undergone positive selection. This is consistent with the results found in the association analysis that multiple sub-regions in the segment contribute to the differences between HB and HQLA. It is hard to discriminate the functional mutations from all SNPs inside each haplotype because they are almost complete linkage disequilibrium across all high-weight chicken. However, candidate intervals have been narrowed to a few Kb levels (lengths range from 2 Kb to 12 Kb, Fig. 3a) and causal mutations are expected to be located within the minimum shared haplotypes present in these breeds.

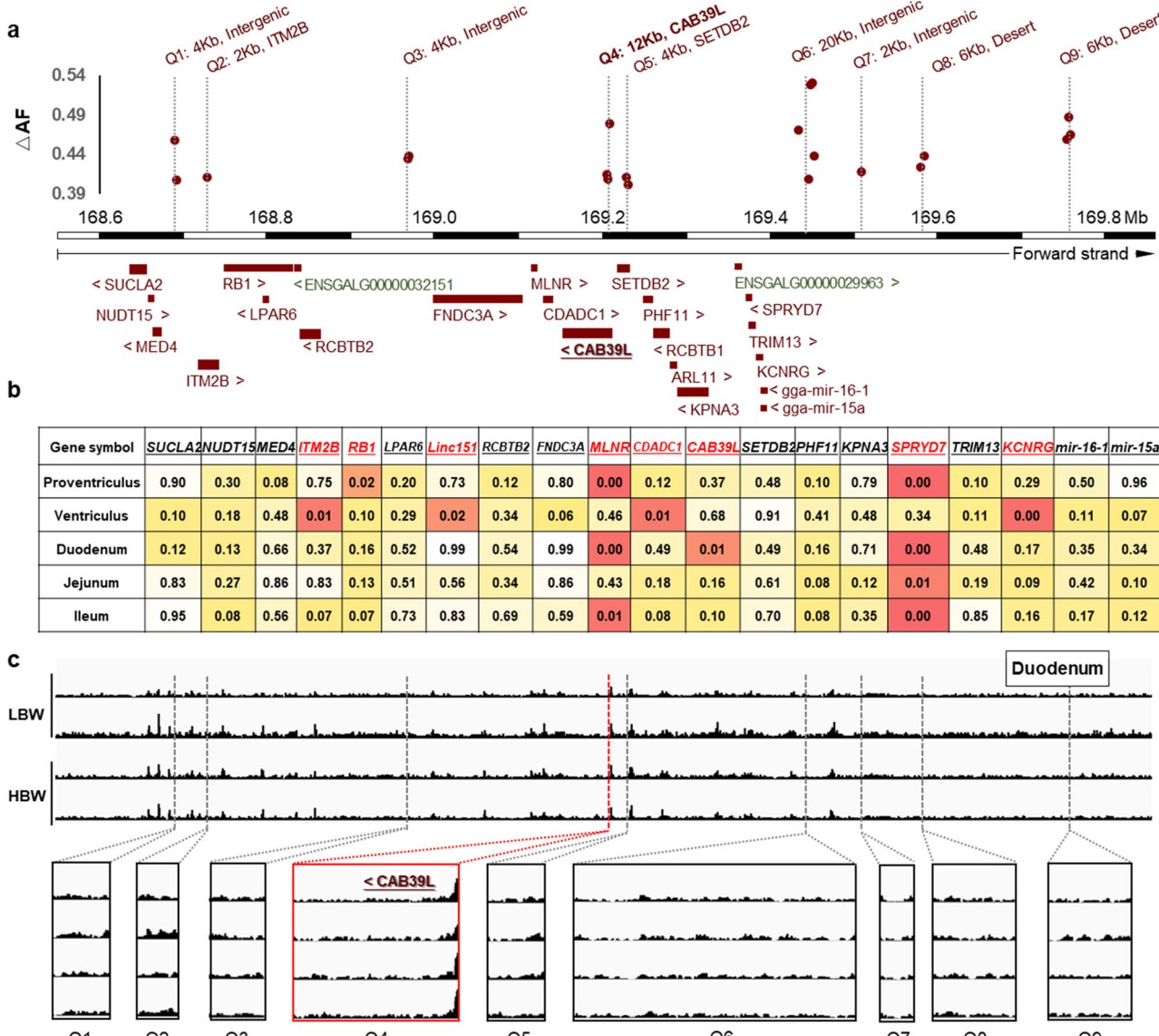

**Fig. 3 Deciphering the genetic architecture, gene expression and regulation mode using multi-population and multi-omics. a** Haplotype block estimation was performed using 16,504 SNPs with 2 Kb bins across the 1.2 Mb candidate region. Nine regions were detected in the HQLA ("Q") haplotype (denoted Q1–Q9) where the 3 sequenced high-weight chicken stocks differentiated from the 7 sequenced low-weight stocks. The ordinate is the haplotype delta allele-frequency difference ($\Delta AF$) between high and low body weight chicken and only bins with $\Delta AF \geq 0.4$ were shown in this figure. The 1.2 Mb segment is illustrated with the locations of these nine regions together with available gene annotations. **b** Results from differential expression analyses of genes detected by quantitative real-time PCR in the 1.2 Mb segments in high ($n = 9$)/low ($n = 10$) 7-week body weight individuals from the $F_{15}$ generation of the deep intercross line in five tissues. Differentially expressed genes ($P$ values < 0.05) highlighted in red. **c** Distriubtion of the chromatin open region in duodenum located the 1.2 Mb and enlarged view of nine Q intervals. LBW represents the two low body weight birds and HBW represents the two high body weight birds. Only one overlapping region (within Q4) which is located upstream of the TSS of *CAB39L* and *SETDB2* genes was founded compared with the haplotype-sharing results.

**Multiple genes are differentially expressed in different tissues.**
All nine sub-haplotypes are located in non-coding regions
(introns or intergenic) which may influence gene expression at
local or long distances and in tissue-specific manners. To examine
the potential functional effect of the variants in these Q-haplo-
types, the expression levels of the 23 genes located in the 1.2 Mb
interval (Fig. 3b and Supplementary Fig. 13) were evaluated using
9 birds with high 7-week weight (BW7: 882.56 ± 18.66 g), and 10
with low 7-week weight (BW7: 717.60 ± 21.15 g) from the $F_{15}$
generation of the deep intercross line. Five digestive tract tissues
(proventriculus, ventriculus, duodenum, jejunum and ileum)
were selected for gene expression analysis detected by quantita-
tive real-time PCR, since phenotypes relating to these tissues were
associated to this region in the GWAS analyses (Supplementary
Tables 3 and 4). Multiple genes were differentially expressed in
different tissues (Fig. 3b and Supplementary Fig. 13) including
*ITM2B* located in sub-region Q2 and *CAB39L* in sub-region Q4.
Outside of these, *MLNR* and *SPRYD7* were more highly expressed
in the high body weight individuals in multiple tissues. These
results suggest that polymorphisms in the selected Q haplotypes
may contribute to chicken weight by altering gene expression in
the same chromosomal segment, perhaps via a network regula-
tion involving multiple target tissues and genes.

**Identification of a regulatory mutation located in the Q4
haplotype.** We use ATAC-seq to identify chromatin accessibility,
which is a crucial component of genome regulation. Here we first
focus on chicken duodenum as an example of in-depth research
because the most significant associations of the evaluated carcass
traits (dressed weight (DW), abdominal fat weight (AFW), evis-
cerated weight (EW) and intestinal length (IL)) in F9 was IL and
duodenum (the first part of small intestine) is known to have
important roles in digestion, appetite regulation and growth[27–29].
We conducted ATAC-seq to profile the accessible chromatin in
duodenum samples from two chickens with high and low body
weights (HBW1, HBW2, LBW1, LBW2) from the $F_{12}$ generation
of the intercross. We obtained 86.9–131.7 million unique mapped
reads and 7,303–29,724 peaks from each sample (Supplementary
Table 6). We assessed the genomic distribution of duodenum
ATAC-seq peaks and found a characteristic enrichment near gene
transcriptional start sites and more intronic and intergenic non-
coding sequences (Fig. 3c, Supplementary Table 7 and Supple-
mentary Fig. 14). Notably, only one overlapping region, which is
located upstream of the TSS of *CAB39L* and *SETDB2* genes
(within Q4 haplotype), was found compared with the haplotype-
sharing results (Figs. 3a and 3c). Within the crucial component of
genome regulation, two tag SNPs (SNP1 chr1:169,208,105 and
SNP2 chr1:169,208,133) were significantly associated with body
weight and other growth traits (GR/CW/IL) in the GWAS ana-
lysis. The QQ (CG) and qq (AA) haplotype sequences from the
open chromatin region for the 2 SNPs
(Chr1:169,207,831–169,208,840; in total 1,010 bp) were cloned in
both orientations (SETDB2(F) and CAB39L(R)) into the pGL3-
basic and pGL3-promoter luciferase reporter vectors. We trans-
fected DF1 cells and measured luciferase activity after 48 h. Fig-
ure 4 shows the comparison of the promoter activity for these,
compared to the promoter-less vector. All fragments were found
to increase luciferase activity compared with empty vector: ~53/
40-fold increase (qq/QQ) in *SETDB2* direction (F) and ~97/218-
fold (qq/QQ) in *CAB39L* direction (R). The luciferase activity was
significantly higher for the QQ constructs than for the qq con-
structs in *CAB39L* direction (R) ($p < 0.001$; ~2.25-fold difference).
For enhancer activity detection, only the QQ constructs showed
significantly increased luciferase activity compared with enhancer
less vector and the corresponding qq constructs (~1.4/1.2-fold

increase in *SETDB2* direction (F)/*CAB39L* direction (R)). This
example shows the Q4 haplotype contains functional element
affecting gene regulation which may affect growth.

## Discussion

Deep intercross-populations between divergent lines is a powerful
experimental design for identifying and fine-mapping chromo-
somal regions (QTL) contributing to complex traits via the
accumulation of recombination events in each generation. A
series of analyses were performed in the $F_2$ and $F_9$ generations of
a cross between native Chinese (HB) and commercial broiler
(HQLA) populations. The results strongly indicated that the
associated region contained multiple segregation haplotypes in
both founder populations harboring regulatory mutations
cumulatively contributing to a major QTL affecting chicken
growth traits.

Earlier works in other chicken populations involving this
region on chromosome 1 have proposed several candidate genes
and/or suggestive functional mutations in the region. A study of
an $F_2$ population identified a 54-bp insertion on *miR-16* that was
significantly associated with increased body weight[15]. This
mutation does, however, not segregate at significantly different
allele frequencies in the HQLA and HB founders of this cross, and
also not in a sample of 50 additional HQLA/HB samples and
other 240 high/low body weight individuals collected for diag-
nostic tests (Supplementary Table 8). Therefore, we consider it a
highly unlikely casual mutation for the differences in body weight
between the founder lines of this study. The situation is similar
for *RB1*, an earlier proposed candidate gene for BW and bone
traits[21] located at around 168.8 Mb on chromosome 1, where the
maximum $|\Delta AF_{(HQLA-HB)}| < 0.4$ in this study. Another associa-
tion detected in this region is that of Yuan et al. reporting the top
association to a SNP (rs13553102) located at upstream of *miR-
15a* (in Block2 of this study) resulting in a significant decrease in
feed conversion ratio[12]. A relatively high $|\Delta AF_{(HQLA-HB)}|$ was
found for this SNP (0.65), but for the other breeds in the low
body weight chicken breed cluster the trend for this poly-
morphism was inconsistent (Supplementary Table 9). A reason-
able explanation for these results is with slightly different sub-
phenotypes of growth traits being the target of selection in dif-
ferent breeds, the preferred polymorphisms in the breeding might
be different to further enhance this trend. The selection of the
target phenotype will make the selection pressure and fitness for
each functional site different, change the genetic architecture
gradually, and further lead to different fine-mapping results.
From another aspect, the presence of multiple, potentially cau-
sative, mutations in this region are coherent with our finding that
multiple sub-haplotypes are likely to have been selected in this
region across the commercial high-weight breeds for multiple
traits including feed conversion ratio, growth rate, meat quality
and so on.

In addition to different selection pressure on alleles affecting
different sub-phenotypes of growth, non-additive genetic archi-
tectures and complex linkage patterns in the founder-populations
used for different breeds are also likely important explanations for
the inconsistent results observed across populations[30]. One of the
most in-depth explorations of growth in chicken have been
performed in the Virginia body weight chicken lines. Starting
with a single significant QTL in the $F_2$ population[31], more than 20
loci have been shown to contribute to long-term responses in
Virginia population, many of which being linked, epistatic and
contributing minor effects from alleles present as standing var-
iations in the base-population[32]. In this study, both the major
QTL interval on GGA1 and the minor QTL region on GGA27
(Fig. 1b) overlapped with QTLs detected in the Virginia lines. But

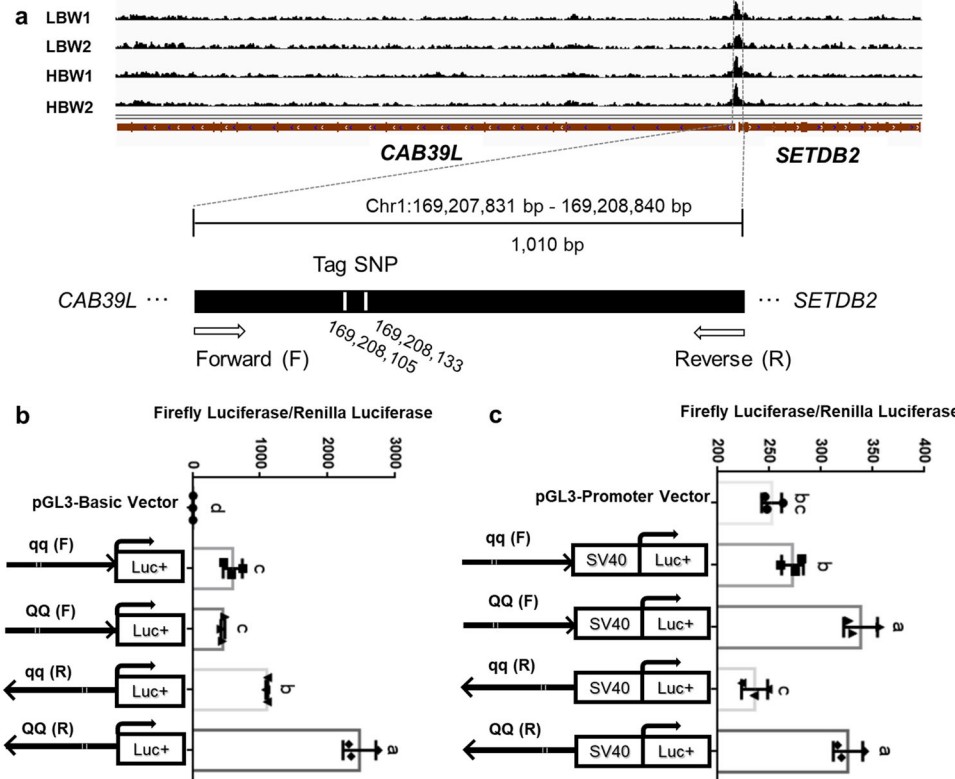

**Fig. 4 Luciferase reporter assay for the open chromatin region in the Q4 haplotype region. a** Schematic representation of the ATAC-seq peak in Q4 haplotype region for two high body weight (HBW1, HBW2) and two low body weight (LBW1, LBW2) chicken duodenums and the target sequence (chr1:169,207,831–169,208,840) inserted into pGL3 vectors. Two genotype DNA fragments (QQ and qq) for the SNPs (chr1:169,208,105 and chr1:169,208,133) are synthesized, except for the tag SNP position, the rest sequences are completely same. The two genotype DNA fragments were cloned to pGL3-Basic/Promoter vectors in the 'Forward (F)' (toward *SETDB2*) and 'Reverse (R)' (toward *CAB39L*) orientation respectively. **b, c** Ratios of firefly to renilla luminescence obtained after transfection of DF1 cells (**b**) with a promoter-less pGL3 vector (pGL3-Basic Vector) and (**c**) with an enhancer-less pGL3 vector (pGL3-Promoter Vector), pairs of sequence-verified preparations of the pGL3 vector including the qq or QQ version of the synthesized fragment cloned either in forward (F) or reverse (R) orientation. ANOVA and multiple comparisons with Least-Significant Difference (LSD) were performed. The data are shown are mean ± SD. Letters indicate significant differences at $p < 0.05$. Values followed by the same letter are not significantly different.

we have not considered the potential impact of high-order genetic interactions (epistasis) on the modeling of genetic associations between sequence-level variations[25,33].

The segregation of multiple haplotypes across the fine-mapped 1.2 Mb segment in the two founder-lines, and the differential expression of multiple genes in this segment between high-weight and low-weight birds, together reflect the complexity in the genetics of growth trait regulation even in a small fragment of the genome. Functional experiment of the open chromatin region located in Q4 haplotype highlighted two SNPs inside a regulatory element as candidate causal mutations and *CAB39L* as a functional candidate gene based on its differential expression in high-weight and low-weight birds. *CAB39L* is a scaffold protein that binds and stabilizes the LKB1 activation loop in a conformation required for phosphorylation of substrates[34]. In particular, its presence enhances the regulatory effect of the pseudokinases, STE20-like kinase family, STRADα and STRADβ, on the activity of LKB1. The AMP-activated-protein-kinase (*AMPK*) is activated upon phosphorylation of Thr172 within the catalytic alpha-subunit by the LKB1–STRAD–CAB39L complex[35–37]. AMPK plays a critical role in hormonal and nutrient-derived anorexigenic and orexigenic signals and in energy balance[38]. Interestingly, selection for muscle growth in pigs resulted in a high frequency of a missense mutation in a muscle-specific isoform of

the AMPK gamma chain[39]. However, the sample size in the differential expression analysis ($n = 19$) is small for this quantitative trait and it can therefore not be expected that other than the largest effects are detected.

Genetic analysis of domestic animals provides a unique opportunity to discover the evolution of genomes under intense selection[40,41]. In the Virginia chicken lines, the large selection responses for body-weight has been shown to be due to a highly polygenic genetic architecture, and where most contributing extended QTL were complex and fine-mapped to multiple linked loci[25,26,42]. Earlier work has shown that most of the selection in that experimental population has been due to selection on standing genetic variation[26]. Here, an evaluation of the region in the ancestral Red junglefowl and other breeds showed that the sub-haplotypes contributing to high weight in the commercial populations were present also in the RJF and several local breeds (Fig. 5a and Supplementary Data 2). According to the theory of genetic hitchhiking, selection on favorable alleles makes them reach high frequencies in a population faster than recombination can degrade the core haplotype around it[43]. Given the recombination rate in chicken[44], it is unlikely that the mosaic pattern observed in the region explored here has been generated during the limited time—10 s or perhaps a few hundred generations—of intense breeding for meat producing chicken breeds. A more

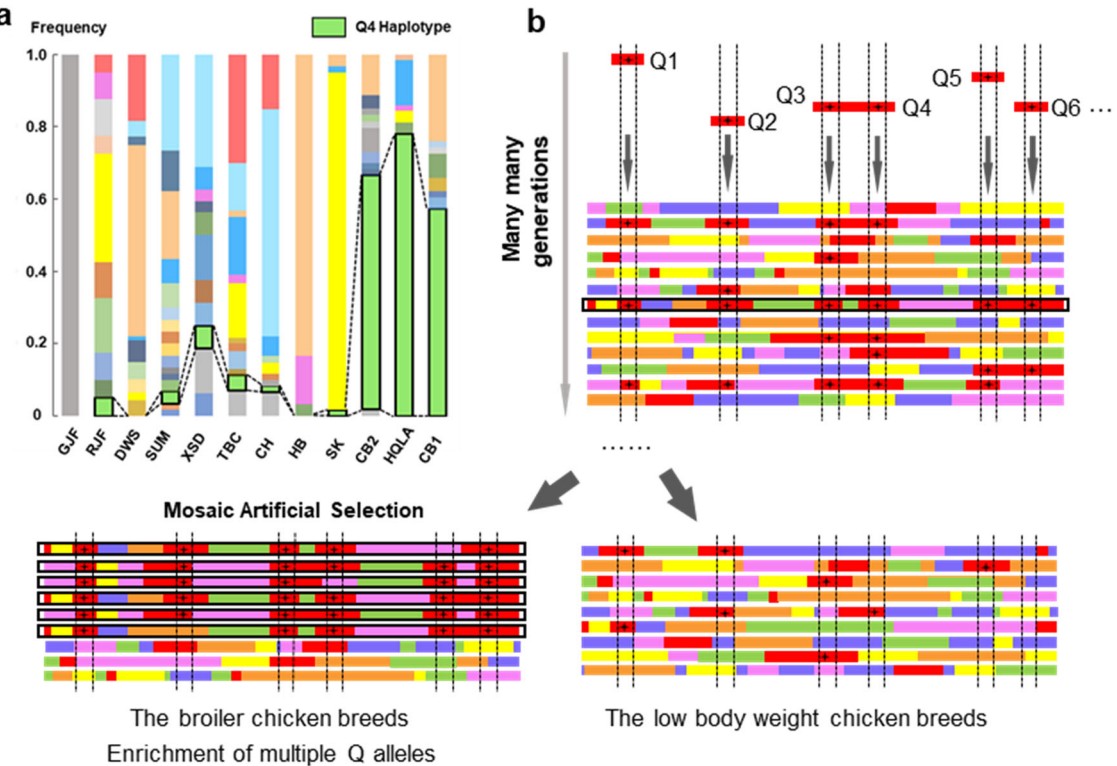

**Fig. 5 Illustration of the proposed model of mosaic artificial haplotype selection for high-weight during chicken domestication.** Our association analyses identified multiple contributing loci in the studied 1.2 Mb region, and segregation of haplotypes with different effects on weight in both the HB and HQLA stocks. **a** Example of haplotype frequency spectrum across 10 domestic, and 2 jungle fowl, stocks for the 12 Kb Q4-region. Each bar represents a breed and the segregating haplotypes in Q4 is assigned a unique color. The HQLA Q-haplotype (green) is present in most populations but in much higher frequencies in the high-weight CB1, CB2, and HQLA populations. **b** We propose that selection for high-weight in the domestic chicken has, in this region, been through mosaic haplotype selection. This by (1) an accumulation of multiple weight-increasing alleles at different times prior, or during, chicken domestication. These provided standing genetic variants, recombined onto multiple genetic backgrounds, as a source of selectable genetic variation. (2) These haplotypes were then, at the onset of modern chicken breeding, accumulated in the breeds developed for different characteristics including high-weight and low-weight. Strong selection, and accumulation of haplotypes with beneficial variants, resulted in a gradual increase in haplotype mosaics containing multiple sub-haplotypes with beneficial alleles.

reasonable explanation is that multiple haplotypes containing different variants across these segments already existed in the population used as founders of the high-weight broiler breeds, to eventually produce the mosaic selection signature observed in this study. Similar patterns have also earlier been discovered in other species[45], and may play an important role in adaptation to diverse habitats for animals and plants[46–48]. In Fig. 5b, we schematically illustrate our hypothesis for how the multiple standing haplotypes across the selected region first emerged, likely by combining standing variants from the red junglefowl during the 1000s of years of chicken domestication, before they were rapidly accumulated in the high-weight chicken breeds during intense artificial selection (Fig. 5b). In this mosaic model, different haplotypes harbor the standing genetic variants and each haplotype does not diversify the population extensively during normal conditions. These can later allow for first rapid and later long-term adaptations by releasing genetic selectable variation by generating new and favored haplotypes via recombination when the population is subjected to intense selection, similar to what has been observed for the Virginia chicken lines[47]. A potentially valuable practical consequence of the results in this study emerges from the observation that all sub-segments in the Q-haplotype have not yet been fixed in all the evaluated broiler breeds (Fig. 5a). This suggests that selection on this major locus is still an on-going event, where different combinations of beneficial variants are selected in the breeds suggesting that there might still be considerable space for improvement by introgression.

In conclusion, the association analyses inferred multiple haplotypes with different effects on weight in chicken, likely due to segregating polymorphisms at multiple, tightly linked regulatory mutations loci in the region. We proposed nine Kb-level candidate segments and selectively validated one of the regulatory roles played in the duodenum. The multiple shorter haplotype genetic architecture of this fine-mapped major QTL region made us propose a mosaic positive selection model consistent with this and earlier findings in chickens. This is an illustration of fine mapping of multiple minor genes in a major QTL. It also provides a new perspective on the genomic consequences of strong artificial selection in chicken, as well as other domestic animals, and new ideas for developing the next generation of genomic selection based breeding.

## Methods

**Experimental population and phenotyping**. A large intercross pedigree was established from two divergent chicken lines, High Quality chicken Line A03 (HQLA), a broiler line bred by Guangdong Wiz Agricultural Science and Technology, Co. (Guangzhou, China), and Huiyang Bearded chicken (HB), a native Chinese breed. In brief, HQLA is a Chinese commercial broiler line founded by the commercial Anak broiler breed and a Chinese indigenous chicken line, followed by strong artificial selection over 10 generations, according to a weight-based selection index. HB is a breed characterized by slow growth and high meat quality. The body weight of HB chickens at 7 weeks of age were, on average, less than one-third of that of chickens from HQLA (Table 1), which is a stock that has been under selection for fast growth for more than 10 generations. Detailed feeding regimes, and $F_0$ to $F_2$ mating schemes, have been described earlier in Sheng et al.[18]. Later AIL generations ($F_3$ to $F_9$) were founded by birds from the $F_2$ population and bred

using random mating. The number of individuals produced in the generations varied from 144 to 294 (Supplementary Table 1).

For the $F_2$ and $F_9$ generations, live body weights were measured in grams at hatching, 2, 4, 6, 8, 10, 12 weeks of age (BW0, BW2, BW4, BW6, BW8, BW10, BW12). Weights at 14 weeks of age (BW14) were measured in $F_9$ only. Growth rates were calculated as the weight gain during the periods 0–4, 4–8, and 8–12 weeks of age (GR0–4, GR4–8, GR8–12). In total, 499 $F_2$ and 602 $F_9$ individuals were slaughtered at week 13 ($F_2$) or 14 ($F_9$). After slaughter, dissections were performed to measure dressed weight (DW), Abdominal fat weight (AFW), eviscerated weight (EW), the combined weight of the ventriculus and the proventriculus (SW, $F_2$ only)[18], all in grams, and intestine length (IL, $F_9$ only) in centimeters. In the $F_9$ individuals, Cholesterol (CHO, mM), Triglyceride (TG, mM) and lactate dehydrogenase (LDH, IU/L) were also measured. Descriptive statistics for these phenotypes were provided in[18] for $F_2$ and in Supplementary Table 2 and Supplementary Data 3 for $F_9$.

**Ethics approval**. All animals used in this study were cared for, and experiments conducted using procedures, that complied with the requirements of the Animal Welfare Committee of Agrobiotechnology of China Agricultural University (approval SKLAB-2014-06-07).

**Whole genome resequencing sample information**. Comparative population genomics involved 253 additional chickens from a range of domestic chicken breeds and two wild junglefowl populations. Analyzed sequences included those from $F_0$ founders for the HQLA-HB AIL (HQLA $n = 16$ and HB $n = 15$), two commercial broiler lines (Cornish $n = 29$, Recessive white Rock $n = 30$), and four Chinese native breeds (Silkies $n = 30$, Chahua chicken $n = 30$, Tibetan chicken $n = 30$, Daweishan Mini chicken $n = 31$; Table 1). In addition, we made use of previously published sequences from the Xishuangbanna game fowl (XSD $n = 8$)[22], Indonesian native breeds (SUM $n = 15$ including Sumatera $n = 5$ and Kedu Hitam $n = 10$)[23] and two jungle fowls (Red jungle fowls; RJF $n = 5$ from the study by ref. [49], and RJFs $n = 5$ and Green jungle fowls; GJFs $n = 9$ from ref. [23]). These chickens were divided into two groups (high and low body weight), with the high-weight group containing the commercial broiler lines (HQLA, CB1, CB2) whose weights were much higher than those of other breeds (Table 1).

**Blood sampling and DNA extraction**. DNA was extracted from EDTA-anticoagulated blood from the wing vein using the phenol-chloroform method ($F_2$) or the Qiagen DNeasy Blood and Tissue Kit ($F_0$, $F_9$ and other breeds) according to the manufacturer's instructions (Qiagen, Hilden, Germany). All animals used in this study were cared for, and experiments conducted using procedures, that complied with the requirements of the Animal Welfare Committee of Agrobiotechnology of China Agricultural University (approval SKLAB-2014-06-07).

**SNP Genotyping in the $F_2$ and $F_9$ populations**. In total, 493 individuals from the $F_2$ generation were genotyped using the Illumina Chicken 60 K SNP Beadchip (Illumina, San Diego, CA), containing in total 57,636 SNPs[50]. SNPs on the sex chromosomes (Z/W), mitochondrial SNPs, and SNPs that could not be assigned to known chromosomes were excluded from the raw data. SNPs that failed to meet the following criteria were removed: individual call rate (>0.9), individual SNP call frequency (>0.9), and minor allele frequency (MAF > 0.05). In total, 493 $F_2$ samples and 41,758 autosomal SNPs were retained for further analysis.

In the $F_9$ generation, double-enzyme digestion genotyping by sequencing (ddGBS) was performed. An *Eco*RI- *Mse*I library was prepared as in ref. [24] and sequencing performed on an Illumina Nextseq500 sequencer. Qualified reads were aligned to the chicken reference genome *Gallus gallus* 5.0 (released 2015) and the TASSEL-4.0 GBS analysis pipeline was used to discover SNPs[51]. The SNP filtering options used in TASSEL are described in detail in ref. [24]. In total, 595 $F_9$ samples and 291,772 SNPs that were evenly distributed across the genome in $F_9$ were retained for further analyses.

**Joint variant calling of new and downloaded data**. Whole genome sequencing was performed for $n = 211$ new samples described above, and sequences from $n = 42$ other birds where downloaded from earlier studies, for a total of $n = 253$ processed samples. Approximately 100 ng of genomic DNA (for each of $n = 211$ samples) was fragmented to a mean size of about 300–400 bp using Covaris E210. The sample preparation workflow was complied with TruSeq Nano DNA Library Prep for NeoPrep Reference Guide (15049722 v01). The quality and concentration of the libraries were assessed with a Qubit2.0 Fluorometer (Thermo, MA, U.S.A.) and an Agilent 2100 Bioanalyzer (Santa Clara, CA, U.S.A.). Paired-end libraries were sequenced on the HiSeq X platform with $2 \times 151$ cycles.

GATK best practices (https://software.broadinstitute.org/gatk/best-practices/) were employed to analyze the $n = 253$ chicken genomes. Prior mapping, adapter sequences were deleted and then the reads which contained more than 50% low quality bases, or more than 5% N bases, were removed. Qualified reads were aligned to the chicken reference genome (Gallus gallus 5.0) with BWA-MEM (version 0.7.10)[52] using '-t 10 -M' as parameters. Initial BAM files were further processed with reordering, sorting and duplicates marking utilizing the Picard

(picard-tools-1.56) package followed by base quality recalibration using the BaseRecalibrator tools in the Genome Analysis Toolkit (GenomeAnalysisTK-3.6, GATK)[53]. Raw variants were called for individual bases using HaplotypeCaller. This process resulted in 253 individual \*.gvcf files which were subjected to joint variant calling using GenotypeGVCFs. The VariantFiltration command was employed to exclude potential false-positive variant calls with the parameter '–filterExpression "QD < 2.0 || FS > 60.0 || MQ < 40.0 || ReadPosRankSum < −8.0" for SNPs and "QD < 2.0 || FS > 60.0 || MQ < 40.0 || ReadPosRankSum < −8.0" for Indels'. Ultimately credible SNPs were identified after using strict filtering criteria with parameters: minor allele frequency <0.05 and call rate >90% in each population. The SNPEff program[54] was used, with the chicken reference genome sequence and GTF annotation files downloaded from Ensembl (version 5.0.89) to annotate variants. In GGA1, structural variations (SV) including deletions (DEL), insertions (INS), inversions (INV) and tandem duplications (DUP) were discovered using the Pindel (V0.2.4) software[55] and copy number variation (CNV) were detected using the CNVnator (V0.3) software[56].

**Genome-wide association analyses**. A mixed linear model (MLM) approach was used for the genome-wide association analyses as implemented in the GCTA package (v1.24)[57]. The MLM model used is described in detail in ref. [58]. The statistical model during analyses of BW2–14 included the sex and batch as discrete covariates and birth weight as quantitative covariate. For DW and EW, sex and batch were included as discrete covariates and BW13/BW14 ($F_2$/$F_9$) as quantitative covariate. For all other traits, only the sex and batch were used as discrete covariates. A quantile-quantile (Q-Q) plot generated in R (v3.0.2) was used to assess the potential impact of population stratification on the genetic association studies (Supplementary Fig. 2). To account for multiple testing across the genome, a Bonferroni correction was applied correcting for the number estimated independent markers from a PCA analysis performed as follows. A subset of SNPs that were in approximate linkage equilibrium with each other was obtained by removing one in each pair of SNPs if the LD was greater than 0.4 using the PLINK v1.07 '--indep-pairwise' command[59]. The squared correlation coefficient ($r^2$) between the genotypes was calculated using the vcftools '--geno-r2' command[60]. Consequently, for the $F_2$ generation, the genome-wide 1% significance threshold was determined as $p$-value $< 8.12 \times 10^{-7}$ (0.01/12,310), and a suggestive association as $4.06 \times 10^{-6}$ (0.05/12,310). For the $F_9$ generation, the thresholds were $1.84 \times 10^{-7}$ (0.01/54,399) and $9.19 \times 10^{-7}$ (0.05/54,399), respectively. The genotype in major QTL on GGA1 (165–175 Mb) of $F_9$ generation was shown in Supplementary Data 4. The GWAS raw data to BW8 in $F_2$ and $F_9$ were shown in Supplementary Data 5 and 6.

**Fine mapping the QTL**. Stage I: The founder-line origin of the chromosomal segments inherited by the $F_9$ individuals were traced back to facilitate identity-by-descent (IBD) mapping of the segment assuming divergent QTL-allele fixation in the founders. The absolute allele frequency differences (ΔAF) between HQLA and HB were calculated for all markers across the segment using the SNPs identified using whole genome resequencing of the $F_0$ founders (see above). A subset of 46 segregating SNPs ($|\Delta AF_{(HQLA-HB)}| \geq 0.95$, i.e., almost completely fixed for alternative alleles in the two lines) from candidate QTL region were selected to discriminate between alleles inherited from HQLA and HB. These were then genotyped on the Fluidigm platform (SNPtype Assays for SNP Genotyping on the 96.96 Dynamic Array IFCs) in 602 $F_9$ individuals using the run thermal cycling protocol 'SNPtype 96×96 v1'. SNPs and individuals were excluded due to bad calling quality (default parameters), individual call rate < 0.9 and SNPs call frequency < 0.9. In total, 31 SNPs and 575 $F_9$ individuals passed this filtering and missing genotypes were imputed and corresponding recombinant IBD blocks deduced using BEAGLE V4.0[61]. For each recombinant block, the distributions of BW8 and BW14 were evaluated in the 72 individuals carrying recombinant segments (in total 144 chromosomes). The significance of weight differences between chromosomal IBD segments of HQLA-origin and HB-origin were calculated using an unpaired $T$-test (Supplementary Data 7 and 8).

Stage II: The two blocks with the strongest associations to weight (blocks 1 and 2; 168.6–169.8 Mb) were subjected to a second round of targeted genotyping in the 602 available individuals from the $F_9$ generation. New SNPs were chosen based on the following three principles. First, to also tag segregating variants in the founders the allele-frequency selection criterion was relaxed to $|\Delta AF_{(HQLA-HB)}| \geq 0.75$ (mean|ΔAF| ± two times of standard deviation in this 1.2 Mb interval). When multiple SNPs met this criterion and were separated by less than 100 bp, all but one were removed. We selected 61 SNPs in this step (including 9 SNPs genotyped earlier using the GBS method). Second, if a detected SNP was predicted as a missense/splice mutation or was located in the 3′/5′ UTR region of a gene, a softer selection criteria was used as $|\Delta AF_{(HQLA-HB)}| \geq 0.3$. 18 SNPs were selected under this condition. Third, 10 SNPs that reached the genome-wide significance in the earlier $F_2$ Chip-GWAS, but were not part of the $F_9$ GBS set were also included. These 89 SNPs were genotyped using matrix-assisted laser desorption/ionization-time of flight technology (MALDI-TOF, Sequenom®). Thirteen SNPs were excluded after quality control due to bad calling, resulting in 76 SNPs being appended to the third GWAS in the same $F_9$ cohort.

**Selective sweep scans**. To investigate the signatures of selection in the three sequenced broilers and other breeds, five statistical tests were used including the frequency spectrum-based Tajima's D, π and haplotype diversity (H) methods, the linkage disequilibrium-based XP-EHH method and the population differentiation-based Fst method. Three different genomic segments were analyzed: (1) GGA1:168.6–169.8 Mb, (2) GGA1: 165–175 Mb, and (3) the whole GGA1. For these three intervals, the Tajima's D statistic was obtained in bins of 250 bp or 25 Kbp. Calculations of Fst between HQLA and HB were done on a per-site basis or in bins of 20 Kbp. The nucleotide diversity (π) was measured in sliding windows, with a window size of 500 bp and a step size of 250 bp or a window size of 50 Kbp and a step size of 25 Kbp. XP-EHH value normalization at each locus between HQLA and HB was estimated by selscan program[62] and the genetic map for our population was 2.8 cM/Mb for GGA1[18]. Haplotype diversity (H) was calculated for each population as described by Nei and Tajima[63]. The H statistic was calculated in bins with 2 Kb.

**Haplotype blocks**. Genome resequencing data was used to construct haplotypes. Eleven populations were divided into a high-weight group (high; 3 broiler populations) and a low-weight group (low; other 8 breeds except GJF). Haplotype block estimation was performed in PLINK with 2 Kb bins across the 1.2 Mb candidate region. The regions defining the Q-haplotype of HQLA was identified using a criterion of $\Delta AF_{Q(High-Low)} \geq 0.4$. If the haplotypes in two consecutive bins both reach this criterion, the windows will be merged to create a longer Q haplotype sub-segment.

**Haplotype-based association analyses**. In total, 100 haplotypes were detected of which 16 existed at a frequency > 0.01 ("Major haplotypes") and the remaining 84 were grouped ("Other haplotypes"; total frequency = 0.19) (Supplementary Data 9). A haplotype-based association analysis was performed in the 1.2 Mb fine-mapped QTL region using model[25]:

$$Y = X\beta + Zu + e$$

Here, $Y$ is a column vector containing the BW8 of the $F_9$ individuals. $X$ is the design matrix including the coding for the sex of the birds. $Z$ is the design matrix for 17 columns containing the each haplo-genotype number (coded as 0,1,2) of each individual. $\beta$ is a vector with the estimate of the fixed effect of sex. $u$ is a column vector with the estimates for allele substitution effects for each haplo-genotype, and $e$ is the normally distributed residual.

**Examination of population structure**. The population-level phylogeny was inferred using the Neighbor–Joining method[64]. Evolutionary analyses were conducted in MEGA7[65] based on independent-pairwise SNPs and the tree is drawn by interactive tree of life (iTOL) online tool[66]. All SNPs were pruned using the indep-pairwise option in PLINK, with a window size of 50 SNPs, a step of 5 SNPs, and $r^2$ threshold of 0.1.

**RNA extraction and real-time PCR assay (Q-PCR)**. Nine birds with high 7-week weight (BW7: 882.56 ± 18.66 g), and 10 with low 7-week weight (BW7: 717.60 ± 21.15 g) from the $F_{15}$ generation of the deep intercross line were used for the quantitative real-time PCR (Supplementary Data 10). Total RNA was extracted using HiPure Universl miRNA Kit (Magen, Beijing, China) followed the manufacturer's instructions. RNA extractions were treated with RNase free DNase I (Magen, Beijing, China) to remove potentially contaminating DNA. RNA was reverse-transcribed into cDNA with PrimeScript™ RT Master Mix (Perfect Real Time) (Takara, Japan) using 1 μg total RNA. The expression of specific gene (except miRNA) was quantified by real-time PCR using Biomark HD System with 96.96 dynamic arrays (Fluidigm Corporation, CA, USA) according to the "Fast Gene Expression Analysis Using EvaGreen on the BioMark™ or BioMark HD System" in the user guide. miRNA was reverse-transcribed into first-strand cDNA with TransScript miRNA First-Strand cDNA Synthesis SuperMix Kit (TransGen Biotech, Beijing, China). The expression of miRNA was quantified by real-time PCR using a Roche LightCycler® 480 instrument with the Roche LightCycler® 480 SYBR Green I Master Mix (Roche Applied Science, Indianapolis City, U.S.A.). gga-U6 was the reference gene for normalization for miRNA and GAPDH (glyceraldehyde 3-phosphate dehydrogenase) was the reference gene for other target genes. For all genes relative expression levels were calculated using the $2^{-\Delta\Delta Ct}$ method. Primers used for Q-PCR are listed in Supplementary Table 10, the universal miRNA qPCR Primer was provided by TransScript miRNA First-Strand cDNA Synthesis SuperMix Kit (TransGen Biotech, Beijing, China).

**Genotyping the miR-15a-16 mutation**. PCR sequencing was performed to identify reported variations of miR-15a-16 in all $F_0$ individuals. PCR amplification primers were forward: tcctcagtaaatacccacata and reverse: gaactgcattaactacaaaatc according the reference[15]. For more confirmation, another 25 unrelated individuals of HQLA and 25 unrelated individuals of HB were used for additional diagnostic tests.

**Luciferase reporter assay**. Both allelic forms (QQ and qq) of the 1010 bp open chromatin fragment were synthesized and cloned into the pGL3-basic and pGL3-promoter luciferase reporter vectors (Promega Corporation). The sequence and orientation of the inserts were confirmed by sequencing. For cell culture, DF1 (a

chicken fibroblast cell line) cells were cultured in 24-well plates with DMEM (Gibico, Carlsbad, CA, USA) supplemented with 10% FBS (Gibico, Carlsbad, CA, USA) in a 37 °C incubator with 5% $CO_2$. Using Lipofectamine 2000 (Invitrogen, Carlsbad, CA) according to the manufacturer's recommendations, we transfected per well cell in a 24 well plate (~80%–90% confluency) with a mixture comprising 720 ng of the pGL3 firefly luciferase reporter construct, 90 ng of the pRL-TK Renilla luciferase construct (Promega Corporation) and 3 μl Lipofectamine 2000. The luciferase assay was performed 48 h after transfection using the Dual Luciferase Reporter Assay system (Promega Corporation) and an Infinite F200 Luminometer (Tecan, Switzerland). Ratios of firefly luminescence/Renilla luminescence were calculated. For each test construct, one expression value was obtained as the average of three technical replicates in each plate (Supplementary Data 11).

**ATAC-Seq**. Duodenum of high and low body weight chickens at 7 weeks ($n = 4$) were used for ATAC-Seq. All experiments were performed based on the method of Jason D. Buenrostro[67]. Sequencing was performed on Illumina HiSeq platform. Clean reads were obtained from the raw reads by removing the adaptor sequences. The clean reads were then aligned using the bwa program. Peak calling was conducted using macs2 software with cutoff $q$ value < 0.05.

**Statistics and reproducibility**. Details number of biological samples or replicates can be found in the figure or figure legends. Statistical testing was performed in R, GraphPad Prism 8 and Excel. Different standards are used to considered as statistically significant for GWAS, gene expression and ATAC peak calling (see the detail method of each section). All analyses are reproducible with access to genetic data (see "Data availability").

**Reporting summary**. Further information on research design is available in the Nature Research Reporting Summary linked to this article.

## Data availability
The new sequence reads have been deposited in the SRA database (SRA accession: PRJNA547951, PRJNA647930). We declare that the data generated in this study are available within the article and its Supplementary Data files.

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

## Acknowledgements

Jiangli Ren, Zhuolin Huang, and Ying Guo are acknowledged for their valuable contributions during DNA Libraries generation. We also thank Da Yang and Yanjun Zan for useful input on the statistical analysis and the manuscript. This study was financially supported by the National Natural Science Foundation of China (NSFC, 31672411, 31961133003 to X.H. and 31902143 to Y.W.), the 948 Program of the Ministry of Agriculture of China (2012-G1[4] to X.H.), the Earmarked Fund for Modern Agri-industry Technology Research System (Grant No. CARS-41 to D.S.).

## Author contributions

X.H., N.L., D.S., and H.Q. initiated the study and designed the project with Ö.C.; C.L. developed, planned and bred the HQLA-HB advanced intercross chicken lines; Y.W., X.C., C.L., C.F., J.L., F.G., and Z.J. designed, planned, bred, bled, phenotyped and extracted DNA; Y.W., X.C., and Z.S. performed the quality control of the genotype data; X.C. performed the gene expression analyses; Ö.C. and Y.Z. designed the statistical analyses; Y.W., C.Z., and C.B. performed the data analyses; Y.W. and Ö.C. summarized the results and wrote the manuscript.

## Competing interests

The authors declare no competing interests.
