## [Peer Review File · Communications Biology]

Reviewers' comments:

Reviewer #1 (Remarks to the Author):

Growth rate is an important character for broiler. This manuscript confirmed a known major effect QTL on Chromosome 1 based on both F2 and F9 intercross populations. An in-depth analysis narrowed down the QTL into 1.2 Mb. Using some extra resequencing data from more breeds, the authors focused on 9 short sub-intervals because they showed high F_{st} between high weight and low weight chicken breeds, then one of the sub loci was verified by some functional experiments in F15 generation. The authors also proposed a hypothesis that multiple haplotypes or sub-loci cumulatively contributed the genetic variance for growth.

The intercross population and the data generated is extensive and potentially impactful as a resource. However, the analysis is not logically and clearly presented. In all I could not be fully convinced by the fine-mapping and functional verification. The major concerns are as follows:

Line, comment

Line 111: Considering 530 individuals carried unrecombined progenitor chromosomes in the 3.1 Mb region, the authors should focus on the dominance-recessiveness relationship, and compute the detailed phenotype scale for the three genotypes, HQLA/HQLA, HQLA/HB, and HB/HB for the 3.1Mb region.

Line 113 and In Fig. 1d, only the first two proximal blocks show significant p value using unpaired T-test, however, the blocks 1,2 and 3 show obvious differences in scales for BW8 between the HQLA- and HB-origins alleles. Because the authors did not show the sample number used for the five blocks, it is not clear if a small sample size limit to compute a significant result for block 3. The total sample size for the 5 blocks are only 72. All of the detailed sample scales should be marked in the Box plots figures.

Therefore, it is not confirmed that the fine mapped interval is just the ~1.2 Mb region (Block1+Block2). More works need to be done to exclude block 3.

Line 135-140: many reasons could lead to a noncontinuous distribution of associated SNPs or selection signals, such as recombined/unlinked regions and variable allele frequency in founders. It lacks adequate evidence to postulate the hypothesis that multiple linked functional mutations exist.

For Supplementary Fig. 5 and 4, It is not clear if any regions across chromosome 1 are under selection. The figures did not show the whole picture in a larger interval. I also do not think the size of bins is proper for the detection of selection signals. 250 bp or 500 bp bin is too small, which contains rare SNPs. Normally, at least 30 SNPs need to be contained in any bins or

windows for selection sweeps.

Line 141: In the section of haplotype association analysis, I could not follow the method and logic to distinguish the population into 100 haplotypes using 8 tag SNPs.

The authors had reported that there was no recombination for the 530 F9 individuals in the full 3.1 Mb region and only one recombination for the 72 F9 individuals in the ~ 1.2 Mb region (Block1+Block2). So I am a little confused that why so many different recombinations exist from F0 to F9 with an allele frequency $\sim 20\%$.

And considering that the "high-weight" HQLA breed should under strong artificial selection for high weight in history, why so many haplotypes exist in the population, without a major haplotypes?

I think the authors should confirm the accuracy of their haplotype genotyping, before they propose the hypothesis at least two loci contributing to the differences in weight. I suggest using more SNPs to trace the haplotypes from F0 to F2, then to F9, and make the fine-mapping more compatible.

Line 167: In the section of "Exploiting haplotype-sharing discovered nine sub-haplotypes", I still do not sure about the method and logic is correct. Sub- haplotypes and sub-loci is two different concepts. Nine high- and low-weight chicken breeds will contain much more haplotypes and recombinations. The high F_{st} regions between high- and low-weight chicken breeds, would be unnecessary to equivalent to the difference of HB and HQLA. The research target is changed.

Line 202: the result of ATAC-seq is not fully present. More information for ATAC-seq need to show, such as the sequencing data and samples, the peak number, the repeatability. Is there any difference between HB and HQLA for the peak in Q4?

Line 203: The authers say they focused on duodenum beaucse the most significant associations of carcass traits in F9 was intestinal length (IL). However, this is no data support it. I checked Supplementary Table 3 and 4, it seems IL is not the most significant. And, I also not sure about why body weight and intestinal length is related with the function of duodenum. The lower intestine mainly absorb nutrients that were not absorbed in the duodenum.

Line 687: "The raw sequence reads have been deposited in the SRA database (SRA accession:

PRJNA547951)“should not belong to Acknowledgements, but all of the related raw data should be put into an independent data section.

Some minor concerns:

Line 30: “we detected multiple sub loci in the QTL region that cumulatively contributed 14.4% of the genetic variance for growth.”Actually, the whole QTL contributed 14.4%, not the sub loci. It will mislead the readers.

In Fig 2C, What is the unit of this effect? Weight(g)?

Line 343: “Comparative population genomics involved 253 additional chickens from a range of domestic”did not belong to the section of “Experimental population and phenotyping”

Line 110: type err for “the s different types”

Reviewer #2 (Remarks to the Author):

The authors have done a nice job in systematically dissecting this major chr1 body weight locus. The authors approach these analyses from many different angles and employ a number of methods, that together amount to a substantial amount of work. These analyses rely heavily on haplotype-based analyses, and given the availability of whole genome sequence data and good pedigree structures, I do wonder what a direct sequence filtering approach, or imputation and association analysis would yield in terms of candidate variants presented. In saying that, the apparent complexity of the region (ie multiple QTL), is likely better served by the haplotype dissections performed, so these methods are fit for purpose.

One assumption made for some of the analyses presented is that the founder lines are fixed for opposite alleles of the QTL(s) of interest. Im uncertain if the author’s previous studies have specifically demonstrated this, however the possibility that the QTLs segregate in both lines potentially undermines some of the analyses. The frequency of the tag SNP rs14917305 in the respective F0 lines for eg might give some indication of breed segregation, however I cant find this info in the paper. It would therefore be good to add the within breed frequencies of the tagSNPs to one of the tables. The assumption that large breeds are fixed (or heavily enriched) for the chr1 Q alleles is another potential source of false negative variant discrimination in the analysis of the 253 WGS animals. The authors plainly state this assumption, though a statement somewhere that acknowledges the potential downside of the approach is warranted.

Overall the paper presents a nice piece of work and is well written, I enjoyed the paper as it

presents a brave analysis of what looks to be a very complicated locus. A few further minor comments, questions, and typos are highlighted below:

- Was the 54bp miR-16 insertion (or known tag variant thereof) physically genotyped in the different populations? What about the rs13553102 variant? Although the allele frequency might not deviate strongly between breeds segregated by bodyweight, it might still have an effect if the founder frequency assumption does not hold true. This would be nice data to include given that the paper by Jia et al (2016) propose a competing hypothesis that that mutation underlies the locus
- The work aimed at identifying eQTL using qPCR is interesting and a nice addition to the paper, however the sample size is borderline to detect anything other than the largest effects, so a caveat on the power of discovery here is probably warranted.
- The functional analysis of the regulatory variants highlighted by ATAC-seq is also a nice piece of work. Are the variants in the proposed regulatory haplotype evolutionarily conserved?
- I liked Figure 5b as support for the 'mosaic standing variation' hypothesis, though it seems unusual to introduce this in the discussion, as opposed to presenting as a concluding statement/reference in the results. It is no big issue but the journal may have a further opinion here re format

Line 47 "for natural variation of human complex traits" statement is repetitive/redundant in this sentence.

Line 49 Delete "making major contributions to complex traits with examples" from this sentence

Line 110 Typo in the bracketed statement

Line 132 Change 'were here more significant than the earlier' to 'were more significant in this analysis than the earlier'

Line 208 Typo: founded>found

Line 211 – how big was the interval that was cloned into pGL3? I see the size in the figure and methods but would be good to state this in this results paragraph also so it is clear

The response to comments from Reviewer 1

Reviewer #1: Growth rate is an important character for broiler. This manuscript confirmed a known major effect QTL on Chromosome 1 based on both F2 and F9 intercross populations. An in-depth analysis narrowed down the QTL into 1.2 Mb. Using some extra resequencing data from more breeds, the authors focused on 9 short sub-intervals because they showed high F_{st} between high weight and low weight chicken breeds, then one of the sub loci was verified by some functional experiments in F15 generation. The authors also proposed a hypothesis that multiple haplotypes or sub-loci cumulatively contributed the genetic variance for growth.

The intercross population and the data generated is extensive and potentially impactful as a resource. However, the analysis is not logically and clearly presented. In all I could not be full convinced by the fine-mapping and functional verification. The major concerns as bellow:

Line, comment

1. Line 111: Considering 530 individuals carried unrecombined progenitor chromosomes in the 3.1 Mb region, the authors should focus on the dominance-recessiveness relationship, and compute the detailed phenotype scale for the three genotypes, HQLA/HQLA, HQLA/HB, and HB/HB for the 3.1Mb region.

Answer 1: Thank you for your useful suggestion. We apologized that we confused the total number of samples between GBS genotyping (n = 595 after quality control from 602 samples) and Fluidigm genotyping (n = 575 after quality control from 602 samples). Thus, the correct total number of individuals carried unrecombined progenitor chromosomes in the 3.1 Mb region was 503 (575-72). We have done this and added new data in new Figure 1d, and added a description of this result. The revealed that the HB allele is partially dominant with, as expected, a BW8 decreasing effect. The following sentences have been added:

Page 8, Line 117, “In total, 503 F9 individuals carried unrecombined progenitor chromosomes (HQLA/HQLA, HQLA/HB, HB/HB) in the 3.1 Mb region and

significant differences in BW8 were detected between individuals carrying the three different founder-breed origin genotypes (Fig 1d)”.

2. Line 113 and In Fig. 1d, only the first two proximal blocks show significant p value using unpaired T-test, however, the blocks 1,2 and 3 show obvious differences in scales for BW8 between the HQLA- and HB-origins alleles. Because the authors did not show the sample number used for the five blocks, it is not clear if a small sample size limit to compute a significant result for block 3. The total sample size for the 5 blocks are only 72. All of the detailed sample scales should be marked in the Box plots figures. Therefore, it is not confirmed that the fine mapped interval is just the ~1.2 Mb region (Block1+Block2). More works need to be done to exclude block 3.

Answer 2: Thank you for your useful suggestion. First, we added the sample size (number of chromatids) for the respective haplotypes to the new Figure 1e.

As shown in the figure, the number of HQLA-alleles in Block3 (n=45) is lower than for Block2 (n=73). To explore this further, a global (3.1 Mb) ancestry analysis was performed to illustrate the recombination events differentiating the five blocks (new Supplementary Fig. 5):

Supplementary Figure 5. Ancestral origin of alleles for the five blocks in the 72 individuals (144 chromatids). Blue/red lines represent chromosome segments of HQLA/HB origin. The difference between Block2 and Block3 are 28 chromatids converted from HQLA to HB.

We computed the detailed phenotype scales for the three genotypes, HQLA/HQLA, HQLA/HB, and HB/HB for each block in 72 individuals (new Figure 1f). The effect distributions for the three genotypes in the first two blocks are HQLA/HQLA > HQLA/HB > HB/HB and thus consistent with what was observed across all 503 individuals (See above and Figure 1d). For Block3, number of HQLA/HQLA individuals (n=3) is too low to estimate a reliable mean value. The mean for individuals with the HB/HB genotype in this block (n=30) increased compared to individuals with the same genotype in Block1 and Block2. The only difference between Block2 and Block3 is the 28 chromatids converted from HQLA to HB (Supplementary figure 5; above). Given this data, we find it inappropriate to conclude anything definite about the effect of HQLA alleles in block 1 or 2 vs block 3. However, the changes in means do suggest that the HB segments in Block 1 and 2 decrease BW more than the HB segments in Block 3 -> 5. This is consistent with the driver of the effect would be located in Block 1 / 2. Further, the mean of the HB/HB

genotype in Block 3 is more similar to those in Block 4 / 5 than Block 1 / 2. That is also consistent with Block 3->5 not containing any functional allele & hence not contributing to the phenotype, whereas Block 1->2 carries a “weight-decreasing” HB allele.

Hence, we interpret the observed differences in BW8 between the HQLA- and HB-origins alleles (different but not significant) as most likely being a consequence of the remaining linkage in the region where 116 (144-28) unrecombined chromatids will contribute to the phenotypes in block 1/2 and also 3. Based on the results from the above analysis, we find it motivated to focus on Block1 and Block2 (168.6–169.8 Mb) in the further analysis.

The following sentences have been added:

Page 8, Line 125, “For each block interval, we also computed the detailed phenotypic scales for the three genotypes and significant differences were observed among the three groups only in the first two proximal blocks ($P < 0.05$, Fig 1f). The differences between Block2 and Block3 were due to 28 HQLA to HB conversions (Supplementary Fig. 5), resulting in a significantly lower BW8 of HQLA/HB and HB/HB individuals in Block1 and Block2 compared to Block3, Block4 and Block5 (HQLA/HQLA not included due to small sample size, Fig 1f)”.

3. Line 135-140: many reasons could lead to a noncontinuous distribution of associated SNPs or selection signals, such as recombined/unlinked regions and variable allele frequency in founders. It lack adequate evidence to postulated the hypothesis that multiple linked functional mutations exist.

For Supplementary Fig. 5 and 4, It is not clear if any regions across chromosome 1 are under selection. The figures did not show the whole picture in larger interval. I also do not think the size of bins is proper for the detection of selection signals. 250 bp or 500 bp bin is too small, which contains rare SNPs. Normally, at least 30 SNPs need to be contained in any bins or windows for selection sweeps.

Answer 3: First, in this case we consider it highly likely that the noncontinuous

distribution of the associated SNPs result from the variable allele frequencies in the founders (new supplementary Fig. 6b). This as the high recombination rate in chicken makes it unlikely that it would result from the mixtures of alleles taking place during the intercrossing from F0 to F9.

Second, it is worth to note that the breeding of HQLA (High Quality chicken Line A) was performed in a closed population founded by intercrossing the commercial Anak broiler breed with a Chinese indigenous chicken line. After this initial cross, strong artificial selection was performed for more than 10 generations using a weight-based selection index. So, the current diversity in the HQLA results from a breeding process where HQLA weight-increasing alleles most likely originates from the parental commercial stock used in the crossing and later the accumulation of these via directed selection.

Third, based on the above a likely assumption is that the high-weight allele (Q) of HQLA in the studied region originates from the Anak broiler, and that the directional selection for weight enriched it. However, the SNP diversity in the associated region is unlikely due to a significant hitchhiking effect from this selection since: (1) the HQLA was founded by a hybridization event, and (2) the selection time since the founder-event is short (~10 generations).

In conclusion, we therefore consider it likely that multiple haplotypes containing different variants across these smaller segments originates from the high-weight population used as founder for the HQLA (the Anca) and that the mosaic selection signature observed in the current study was due to ancestral haplotype sharing in the admixed population of commercial high-body weight breeds of which the Anca is one.

The following sentences have been added:

Page 9, Line 151, “It is considered unlikely that the noncontinuous association signal observed was due to a recombination breakup of founder haplotypes in the AIL due to

the limited number of generations of intercrossing (F0 to F9) and the recombination rate in chicken. Instead, it is considered more likely to have arisen from variable allele frequencies of the SNPs in the founders (Supplementary Fig. 6). No strong positive selection signal was found using standard measures of the genetic diversity ($\Delta AF/\pi$ /haplotype diversity (H)/Fst/XP-EHH; Supplementary Figs. 6, 7, 8 and 9). These results are consistent with the breeding history of the commercial HQLA stock (details in Materials section), that was formed by first crossing two divergent but outbred population (increasing the haplotype diversity) followed by strong directional selection for increased weight over 10 generations (likely resulting in selection signatures due to selection of longer old haplotypes than creation of new via genetic hitchhiking across multiple individually contributing SNPs at candidate functional loci). This distinct mosaic pattern observed is therefore different from the genetic architecture expected around a strongly selected single causative mutation, making it difficult to exclude any significant SNPs as tags of contributing variants”.

Further, we appreciate the feedback and suggestion about the implementation of selection detection. The analyses were performed using modified settings and new selection sweep results obtained by calculating Fst between HQLA and HB in sliding windows (window size 50 Kbp, step size of 25 Kbp) are now reported in new Supplementary Figures 6a, 7 and 8. We did not observe any selection signal in this QTL region using these settings, which is consistent with our above analysis to suggest little hitchhiking for SNPs near causative mutations.

4. Line 141: In the section of haplotype association analysis, I could not follow the method and logic to distinguish the population into 100 haplotypes using 8 tag SNPs. The authors had reported that there was no recombination for the 530 F9 individuals in the full 3.1 Mb region and only one recombination for the 72 F9 individuals in the ~1.2 Mb region (Block1+Block2). So I am a little confused that why so many different recombinations exist from F0 to F9 with an allele frequency ~20%.

Answer 4a: We realize now that the logic of “recombination” here was not

sufficiently clear. In the previous analyses, the focus was on inferring ancestry information (IBD) based on differences between the two crossed breeds. Therefore, we conducted the analysis in the population using between-population fixed SNPs (fixed in 16 HQLA vs 16 HB). In this section, the haplotypes were determined on an individual basis (across the 32 samples), providing more detailed insights to haplotype differences present between individuals also within population. These within population segregating haplotypes do not only highlight recombination events from F0 to F9, but also reflect the haplotype diversity of the F0 individuals themselves. For example, Figure 3 illustrates that 5 haplotypes (MAF>0.01) segregate in HQLA and 3 (MAF>0.01) in HB.

The following sentences have been changed:

Page7, Line105, “To trace haplotypes inherited from the founder breeds through the experimental cross, we first used founder-discriminatory markers (breed-level) to perform an identical-by-descent (IBD) analysis from F0 to F9.”

Page9, Line138, “In an attempt to utilize individual-level haplotype diversity in the founders to fine-map the region further...”

And considering that the “high-weight” HQLA breed should under strong artificial selection for high weight in history, why so many haplotypes exist in the population, without a major haplotypes?

Answer 4b: As described in the responses to the earlier questions, the diversity of the HQLA is consistent with the HQLA being a commercial stock formed by recent divergent crossing with subsequent directional selection.

I think the authors should confirm the accuracy of their haplotype genotyping, before they propose the hypothesis at least two loci contributing to the differences in weight. I suggest using more SNPs to trace the haplotypes from F0 to F2, then to F9, and make the fine-mapping more compatible.

Answer 4c: The accuracy of the genotyping has been evaluated in our previous study [1]. By validation using the SNP chip, we found that the consistency between this GBS genotyping result and the SNP chip is over 99% and based on this expect the genotyping should be reliable. To avoid introducing haplotype phasing errors due to the considerable allele frequency fluctuations and the haplotype diversity present in this region, only haplotypes with $MAF > 0.01$ were considered. Together, we consider these as contributing to the reliability of the conclusions. Unfortunately, GBS data is not available for the F2 making it difficult to track the inheritance of haplotypes from F0->F2->F9, however as almost all F0 haplotypes were reproduced in F9, this provides a further illustration of the credibility of our results.

Reference:

[1] Wang, Y. et al. Optimized double-digest genotyping by sequencing (ddGBS) method with high-density SNP markers and high genotyping accuracy for chickens. PLoS One 12, e0179073 (2017).

5. Line 167: In the section of “Exploiting haplotype-sharing discovered nine sub-haplotypes”, I still do not sure about the method and logic is correct. Sub-haplotypes and sub-loci is two different concepts. Nine high- and low-weight chicken breeds will contain much more haplotypes and recombinations. The high F_{st} regions between high- and low-weight chicken breeds, would be unnecessary to equivalent to the difference of HB and HQLA. The research target is changed.

Answer 5: The purpose of this section is to describe an approach for further analyzing the region to detect possible causal mutations. This under the assumption that high-weight chicken commercial chicken stocks, due to common admixtures in the history of the world-wide chicken population and use of the same founder populations more recently, are likely to share common standing genetic variants with beneficial effects on growth (“Q alleles”), while low-weight chicken breeds instead share the genetic variants with smaller contributions to growth (“q alleles”). Due to more generations of recombination and selection in separated populations, they are more

likely to share haplotypes around causal loci and historical recombinations will contribute information to distinguish between regions more likely to contain true contributing mutations and regions where associations are due to other population genetics factors. In fact, this is a method to expand the evaluation beyond HB/HQLA and use the historical relatedness and similar selection schemes in the chicken for independent and complementary verification of the association results from the divergent cross. Earlier studies have used analyses based on similar assumptions and methods to fine-map the causative mutations [1-3] and the approach used here is an extension adapted to the particular populations analyzed.

References:

- [1] Van Laere, A.S. et al. A regulatory mutation in IGF2 causes a major QTL effect on muscle growth in the pig. *Nature* 425, 832-6 (2003).
- [2] Latifa Karim, et al. (2011) Variants modulating the expression of a chromosome domain encompassing PLAG1 influence bovine stature. *Nature Genetics*: doi:10.1038/ng.814
- [3] Ren J, Duan Y, Qiao R, Yao F, Zhang Z, et al. (2011) A Missense Mutation in PPARD Causes a Major QTL Effect on Ear Size in Pigs. *PLoS Genet* 7(5):e1002043. doi:10.1371/journal.pgen.1002043

6. Line 202: the result of ATAC-seq is not fully present. More information for ATAC-seq need to show, such as the sequencing data and samples, the peak number, the repeatability. Is there any difference between HB and HQLA for the peak in Q4?

Answer 6: We have now added the information for the ATAC sequencing information in Supplementary Table 7 & 8, and Supplementary Fig. 14. We do not know whether there is a difference between HB and HQLA. This as we used samples from high- and low- body weight birds (HBW and LBW) from the F12 generation, rather than HB and HQLA samples. There was no difference in the peak signal of HBW and LBW samples in this open region of chromatin. The following sentences have been added:

Page 13, Line 235, "We conducted ATAC-seq to profile the accessible chromatin in duodenum samples from two chickens with high and low body weights (HBW1,

HBW2, LBW1, LBW2) from the F₁₂ generation of the intercross. We obtained 86.85-131.69 million unique mapped reads and 7,303-29,724 peaks from each sample (Supplementary Table 7). We assessed the genomic distribution of duodenum ATAC-seq peaks and found a characteristic enrichment near gene transcriptional start sites and more intronic and intergenic non-coding sequences (Fig. 3c, Supplementary Table 8 and Supplementary Fig. 14).”

7. Line 203: The authors say they focused on duodenum because the most significant associations of carcass traits in F9 was intestinal length (IL). However, this is no data support it. I checked Supplementary Table 3 and 4, it seems IL is not the most significant. And, I also not sure about why body weight and intestinal length is related with the function of duodenum. The lower intestine mainly absorb nutrients that were not absorbed in the duodenum.

Answer 7: Here we focus on chicken duodenum because it was the most significant trait of the evaluated carcass traits (dressed weight (DW), abdominal fat weight (AFW), eviscerated weight (EW) intestinal length (IL)) in the F9. This is now clarified in the text:

Page 13, Line 231, “Here we focus on chicken duodenum for several reasons. This because the most significant associations of the evaluated carcass traits (dressed weight (DW), abdominal fat weight (AFW), eviscerated weight (EW) and intestinal length (IL)) in F9 was IL and also since duodenum (the first part of small intestine) is known to have important roles in digestion, appetite regulation and growth^{27-29,}”

In addition to this, there are a few other reasons for us to focus on IL:

– Body weight is a highly complex phenotype where multiple physiological processes involved in, for example, bone size, muscle and fat weight, digestive ability etc contribute and likely have different genetic underpinnings. We cannot directly establish the correlation between genes and body weight. By shifting focus from body weight only to also measure sub-phenotypes contributing to body weight, the series of carcass traits we measured in F9 to assist the attempt to better understand the

underlying physiological processes. Since IL had the most significant GWAS signal of the carcass traits, had the same top SNP as the body weight association in multiple periods and displayed positive correlations with body weights measured over time we consider it warranted to further explore this QTL functionally by studying intestinal development and function.

– It can be hypothesized that an increased duodenum length, via an enlarged surface area of the intestine and increased time that food stays in the intestine, might improve food digestion and nutrient absorption. Therefore we used the IL as a morphological intestine phenotype.

– Digestion is important for breaking down food into nutrients, which the body uses for energy, growth, and cell repair. The process of chemical digestion begins in the stomach and continues in the duodenum (the first part of intestine) as pancreatic enzymes and bile are mixed with the chyme to be further broken down so that nutrients can easily be absorbed [1-3]. Although most nutrient absorption occurs further down in the small intestine, the duodenum still contributes to the absorption of vital nutrients, in particular iron and calcium [4-7].

- Enteroendocrine cells (EECs) are found scattered throughout the epithelium of the gastrointestinal (GI) tract from the stomach to the rectum. EECs produce a range of gut hormones that have key roles in the coordination of food digestion and absorption, insulin secretion and appetite and these gut hormones have multifaceted beneficial effects on food intake and body weight [8-11]. Duodenum is the first part of the intestine, in the duodenum large numbers of EECs produce glucose-dependent insulinotropic polypeptide (GIP), cholecystokinin (CCK), and secretin (SCT) [12], and other hormones such as ghrelin, SST, motilin and 5-hydroxytryptamine (5-HT) [11, 13, 14]. 5-HT and ghrelin, for example, affect the appetite, motility, fluid secretion, release of digestive enzymes and body weight [14-16].

References :

- [1]. Biagioli, M. and A. Carino, Signaling from Intestine to the Host: How Bile Acids Regulate Intestinal and Liver Immunity. *Handb Exp Pharmacol*, 2019. 256: p. 95-108.
- [2]. Weiss, F.U., W. Halangk and M.M. Lerch, New advances in pancreatic cell physiology and

- pathophysiology. *Best Pract Res Clin Gastroenterol*, 2008. 22(1): p. 3-15.
- [3]. Schulze, K., Imaging and modelling of digestion in the stomach and the duodenum. *Neurogastroenterol Motil*, 2006. 18(3): p. 172-83.
- [4]. Sato, M., et al., Increased Duodenal Iron Absorption through Upregulation of Ferroportin 1 due to the Decrement in Serum Hepcidin in Patients with Chronic Hepatitis C. *Can J Gastroenterol Hepatol*, 2018. 2018: p. 2154361.
- [5]. Pantopoulos, K., et al., Mechanisms of mammalian iron homeostasis. *Biochemistry*, 2012. 51(29): p. 5705-24.
- [6]. Drakesmith, H., E. Nemeth and T. Ganz, Ironing out Ferroportin. *Cell Metab*, 2015. 22(5): p. 777-87.
- [7]. Diaz, D.B.G., S. Guizzardi and D.T.N. Tolosa, Molecular aspects of intestinal calcium absorption. *World J Gastroenterol*, 2015. 21(23): p. 7142-54.
- [8]. Gribble, F.M. and F. Reimann, Function and mechanisms of enteroendocrine cells and gut hormones in metabolism. *Nat Rev Endocrinol*, 2019. 15(4): p. 226-237.
- [9]. Andersen, A., et al., Glucagon-like peptide 1 in health and disease. *Nat Rev Endocrinol*, 2018. 14(7): p. 390-403.
- [10]. Gribble, F.M. and F. Reimann, Enteroendocrine Cells: Chemosensors in the Intestinal Epithelium. *Annu Rev Physiol*, 2016. 78: p. 277-99.
- [11]. Stengel, A. and Y. Tache, Regulation of food intake: the gastric X/A-like endocrine cell in the spotlight. *Curr Gastroenterol Rep*, 2009. 11(6): p. 448-54.
- [12]. Sjolund, K., et al., Endocrine cells in human intestine: an immunocytochemical study. *Gastroenterology*, 1983. 85(5): p. 1120-30.
- [13]. Itoh, Z., Motilin and clinical application. *Peptides*, 1997. 18(4): p. 593-608.
- [14]. Diwakarla, S., et al., Heterogeneity of enterochromaffin cells within the gastrointestinal tract. *Neurogastroenterol Motil*, 2017. 29(6).
- [15]. Castaneda, T.R., et al., Ghrelin in the regulation of body weight and metabolism. *Front Neuroendocrinol*, 2010. 31(1): p. 44-60.
- [16]. Anderberg, R.H., et al., Glucagon-Like Peptide 1 and Its Analogs Act in the Dorsal Raphe and Modulate Central Serotonin to Reduce Appetite and Body Weight. *Diabetes*, 2017. 66(4): p. 1062-1073.

8. Line 687: “The raw sequence reads have been deposited in the SRA database (SRA accession: PRJNA547951)” should not belong to Acknowledgements, but all of the related raw data should be put into an independent data section.

Answer 8: We have moved this sentence to the new “Data availability” section.

Some minor concerns:

9. Line 30: “we detected multiple sub loci in the QTL region that cumulatively contributed 14.4% of the genetic variance for growth”. Actually, the whole QTL

contributed 14.4%, not the sub loci. It will mislead the readers.

Answer 9: This has been updated to:

Page 2, Line 30, “We detected this QTL that, in total, contributed 14.4% of the genetic variance for growth”.

10. In Fig 2C, what is the unit of this effect? Weight(g)?

Answer 10: We have added “Weight (g)” to the new Figure 2c.

11. Line 343: “Comparative population genomics involved 253 additional chickens from a range of domestic” did not belong to the section of “Experimental population and phenotyping”

Answer 11: We added a new subtitle “Whole genome resequencing sample information” to this paragraph.

12. Line 110: type err for “the s different types”

Answer 12: This has been corrected.

Reviewer #2 (Remarks to the Author):

13. The authors have done a nice job in systematically dissecting this major chr1 body weight locus. The authors approach these analyses from many different angles and employ a number of methods, that together amount to a substantial amount of work. These analyses rely heavily on haplotype-based analyses, and given the availability of whole genome sequence data and good pedigree structures, I do wonder what a direct sequence filtering approach, or imputation and association analysis would yield in terms of candidate variants presented. In saying that, the apparent complexity of the region (ie multiple QTL), is likely better served by the haplotype dissections performed, so these methods are fit for purpose.

Answer 13: Thank you for your positive comments.

14. One assumption made for some of the analyses presented is that the founder lines are fixed for opposite alleles of the QTL(s) of interest. I'm uncertain if the author's previous studies have specifically demonstrated this, however the possibility that the QTLs segregate in both lines potentially undermines some of the analyses. The frequency of the tag SNP rs14917305 in the respective F0 lines for eg might give some indication of breed segregation, however I can't find this info in the paper. It would therefore be good to add the within breed frequencies of the tagSNPs to one of the tables. The assumption that large breeds are fixed (or heavily enriched) for the chr1 Q alleles is another potential source of false negative variant discrimination in the analysis of the 253 WGS animals. The authors plainly state this assumption, though a statement somewhere that acknowledges the potential downside of the approach is warranted.

Answer 14: A new Table (Table 2) has been added to show the within breed frequencies of the top significant SNPs for all traits. Further, we also plotted the relationship between the delta allele-frequency differences between the HQLA and HB of all SNPs in this QTL and their respective P-values (new Supplementary Fig. 3). A positive correlation ($r=0.68$) existed between the allele-frequency differences and P-values, consistent with i) high/low-weight alleles being more likely in HB/HQLA, respectively and ii) the association analysis being able to detect loci here allele-frequencies are different (not necessarily fixed) between the lines.

The following sentences have been added to clarify this point:

Page 7, Line 101, “The GWAS P-values and the difference in allele-frequency between HQLA and HB at the significant SNPs ($\Delta AF(HQLA-HB)$) were highly correlated ($r=0.68$, Supplementary Fig. 3). This correlation is consistent with the basic assumption in our analyses that, at many loci, alleles with significantly different effects on growth (growth-increasing Q in HQLA and growth-decreasing q in HB)

were present in considerably different frequencies in these two phenotypically divergent populations.”

We agree that the analysis of the 253 WGS animals has a problem with false negatives. To address, the following statement has been added in the discussion:

Page 11, Line 195, “We here assumed that one or more of the multiple favorable regions of the large-effect, multi-locus QTL haplotype identified in the cross between HB \times HQLA (the “Q-haplotype”) would also have been selected in other high-weight breeds and that these breeds would therefore, at least in parts, share the Q-haplotype in the region. However, this analysis is also a potential source of false negative variant discrimination”.”

15. Overall the paper presents a nice piece of work and is well written, I enjoyed the paper as it presents a brave analysis of what looks to be a very complicated locus. A few further minor comments, questions, and typos are highlighted below:

Was the 54bp miR-16 insertion (or known tag variant thereof) physically genotyped in the different populations? What about the rs13553102 variant? Although the allele frequency might not deviate strongly between breeds segregated by bodyweight, it might still have an effect if the founder frequency assumption does not hold true. This would be nice data to include given that the paper by Jia et al (2016) propose a competing hypothesis that that mutation underlies the locus.

Answer 15: We used WGS sequencing data to analyze the 54bp miR-16 insertion and

the result is now provided in new Supplementary Table 9. Overall, the mutation frequency spectrum is not consistent with the hypothesis and similar is true for the rs13553102 variant (details in new Supplementary Table 10). Unfortunately, we have no suitable data to explore this further to, for example, show whether this mutation has not been strongly selected in these populations due to it being a functional mutation or not.

16. The work aimed at identifying eQTL using qPCR is interesting and a nice addition to the paper, however the sample size is borderline to detect anything other than the largest effects, so a caveat on the power of discovery here is probably warranted.

Answer 16: Following this suggestion, we have added the following to the revision: Page 17, Line 314, “However, the sample size in the differential expression analysis (n=19) is small for this quantitative trait and it can therefore not be expected that other than the largest effects are detected.” to the discussion.

17. The functional analysis of the regulatory variants highlighted by ATAC-seq is also a nice piece of work. Are the variants in the proposed regulatory haplotype evolutionarily conserved?

Answer 18: We analyzed the sequence conservation using data from 9 species including Human, Mouse, Chicken, Duck, Turkey, Zebra Finch, Flycatcher, Chinese softshell turtle and Anole lizard to evaluate this. This showed that this 1Kb sequence only exists, in whole or in part, in four of these (Chicken, Flycatcher, Turkey, Zebra Finch; detailed in the figure below). Based on this, we would consider it as an independently evolved genome sequence.

18. I liked Figure 5b as support for the ‘mosaic standing variation’ hypothesis, though it seems unusual to introduce this in the discussion, as opposed to presenting as a concluding statement/reference in the results. It is no big issue but the journal may have a further opinion here re format.

Answer 18: We have considered this again and do think it is prudent to put this as part of the discussion. Although we believe we have sufficient reasons to propose this hypothesis, the most direct evidence to support it need to come from resequencing data of commercial broilers hundreds of generations ago for which data, however, is not available. We have now, in light of this comment and as an attempt to improve the logic of the article, included statements referring to this mosaic formation mechanism to the Results “Identifying a mosaic pattern of the 1.2 Mb QTL interval by association analyses to markers segregating in the founder populations”. We are, however, open to move this section based on editorial advice from the journal.

19. Line 47 “for natural variation of human complex traits” statement is repetitive/redundant in this sentence.

Answer 19: This has been deleted.

20. Line 49 Delete “making major contributions to complex traits with examples” from this sentence.

Answer 19: This has been modified.

21. Line 110 Typo in the bracketed statement

Answer 20: This has been corrected.

22. Line 132 Change ‘were here more significant than the earlier’ to ‘were more significant in this analysis than the earlier’

Answer 21: This has been implemented.

23. Line 208 Typo: founded>found

Answer 23: This has been corrected.

24. Line 211 – how big was the interval that was cloned into pGL3? I see the size in the figure and methods but would be good to state this in this results paragraph also so it is clear

Answer 24: This information is now provided as:

Page 14, Line 245, “The QQ (CG) and qq (AA) haplotype sequences from the open chromatin region for the 2 SNPs (Chr1:169,207,831-169,208,840; in total 1,010 bp) were cloned in both orientations (SETDB2(F) and CAB39L(R)) into the pGL3-basic and pGL3-promoter luciferase reporter vectors.”.

REVIEWERS' COMMENTS:

Reviewer #1 (Remarks to the Author):

I appreciate the efforts of the authors. Based on the new analysis from the authors, my doubts for the QTL region narrowed down from ~3 Mb to 1.2 Mb have been resolved. However, the results from the fine-mapping to nine sub-haplotypes and finally got one causative gene and one causative regulation region were not very convincing. Considering that multiple genes are differentially expressed in different tissues and the ATAC-seq can only prove that one of the nine sub-haplotypes may have regulatory functions in the duodenum, whereas other sub-haplotypes cannot be excluded. The author should tone down the finding in the fine-mapping parts.

Reviewer #2 (Remarks to the Author):

I have no further comments/queries

The response to comments from Reviewer 1

Reviewer #1 (Remarks to the Author):

I appreciate the efforts of the authors. Based on the new analysis from the authors, my doubts for the QTL region narrowed down from ~3 Mb to 1.2 Mb have been resolved. However, the results from the fine-mapping to nine sub-haplotypes and finally got one causative gene and one causative regulation region were not very convincing. Considering that multiple genes are differentially expressed in different tissues and the ATAC-seq can only prove that one of the nine sub-haplotypes may have regulatory functions in the duodenum, whereas other sub-haplotypes cannot be excluded. The author should tone down the finding in the fine-mapping parts.

Response: We totally agree with the reviewer. In fact, the conclusion of this manuscript is not the only one causative gene and one causative regulation region. On the contrary, we have repeatedly demonstrated that there may be multiple causative mutations in this region, for example:

P2, Line31, in Abstract: “nine mosaic precise intervals (Kb level) which contain ancestral regulatory variants were fine-mapped and **we chose one of them** to demonstrate the key regulatory role in the duodenum”.

P10, Line167, “Haplotype association analysis suggests **multiple contributing loci** in the 1.2 Mb region”;

P11, Line193, “Exploiting haplotype-sharing discovered **nine sub-haplotypes as prime causal candidates**”;

P13, Line224, “These results suggest that polymorphisms in the selected **Q haplotypes may contribute to chicken weight by altering gene expression in the same chromosomal segment, perhaps via a network regulation involving multiple target tissues and genes**”.

The main reason for the reviewers’ doubts may be that we only analyzed the regulatory mutations in only one tissue (duodenum). We considered this is just an example of multiple regulatory mutations, and does not mean that there is only one causative mutation. We modified the sentence to better illustrate this point:

P13, Line231, “Here we first focus on chicken duodenum as an example of in-depth research because...”